# Parameterizing cloud top effective radii from satellite retrieved values, accounting for vertical photon transport: Quantification and correction of the resulting bias in droplet concentration and liquid water path retrievals.

Daniel P. Grosvenor[1,2], Odran Sourdeval[3], and Robert Wood[4]

[1]School of Earth and Environment, University of Leeds, Leeds, LS2 9JT, UK
[2]National Centre for Atmospheric Science (NCAS), University of Leeds, Leeds, LS2 9JT, UK
[3]Leipzig Institute for Meteorology, Universität Leipzig, Germany
[4]Department of Atmospheric Sciences, University of Washington, Seattle, USA

*Correspondence to:* D. P. Grosvenor
(daniel.p.grosvenor@gmail.com)

**Abstract.** Droplet concentration ($N_d$) and liquid water path (LWP) retrievals from passive satellite retrievals of cloud optical depth ($\tau$) and effective radius ($r_e$) usually assume the model of an idealised cloud in which the liquid water content (LWC) increases linearly between cloud base and cloud top (i.e., at a fixed fraction of the adiabatic LWC). Generally it is assumed that the retrieved $r_e$ value is that at the top of the cloud. In reality, barring $r_e$ retrieval biases due to cloud heterogeneity, etc., the retrieved $r_e$ is representative of smaller values that occur lower down in the cloud due to the vertical penetration of photons at the shortwave infra-red wavelengths used to retrieve $r_e$. This inconsistency will cause an overestimate of $N_d$ and an underestimate of LWP (referred to here as the "penetration depth bias"), which this paper quantifies via a parameterization of the cloud top $r_e$ as a function of the retrieved $r_e$ and $\tau$. Here we estimate the relative $r_e$ underestimate for a range of idealised modelled adiabatic clouds using bispectral retrievals and plane-parallel radiative transfer. We find a tight relationship between $g_{re} = r_e^{cloudtop}/r_e^{retrieved}$ and $\tau$ and that a 1-D relationship approximates the modelled data well. Using this relationship we find that $g_{re}$ values and hence $N_d$ and LWP biases are higher for the 2.1 μm channel $r_e$ retrieval ($r_{e2.1}$) compared to the 3.7 μm one ($r_{e3.7}$). The theoretical bias in the retrieved $N_d$ is very large for optically thin clouds, but rapidly reduces as cloud thickness increases. However, it remains above 20 % for $\tau <$19.8 and $\tau <$7.7 for $r_{e2.1}$ and $r_{e3.7}$, respectively. We also provide a parameterization of penetration depth in terms of the optical depth below cloud top ($d\tau$) for which the retrieved $r_e$ is likely to be representative.

The magnitude of the $N_d$ and LWP biases for climatological data sets is estimated globally using one year of daily MODIS (MODerate Imaging Spectroradiometer) data. Screening criteria are applied that are consistent with those required to help ensure accurate $N_d$ and LWP retrievals. The results show that the SE Atlantic, SE Pacific and Californian stratocumulus regions produce fairly large overestimates due to the penetration depth bias with mean biases of 32–35 % for $r_{e2.1}$ and 15–17 % for $r_{e3.7}$. For the other stratocumulus regions examined the errors are smaller (24–28 % for $r_{e2.1}$ and 10–12 % for $r_{e3.7}$). Significant time variability in the percentage errors is also found with regional mean standard deviations of 19–37 % of the

regional mean percentage error for $r_{e2.1}$ and 32–56 % for $r_{e3.7}$. This shows that it is important to apply a daily correction to $N_d$ for the penetration depth error rather than a time–mean correction when examining daily data. We also examine the seasonal variation of the bias and find that the biases in the SE Atlantic, SE Pacific and Californian stratocumulus regions exhibit the most seasonality with the largest errors occurring in the December, January, February (DJF) season. LWP biases are smaller in magnitude than those for $N_d$ (-8 to -11 % for $r_{e2.1}$ and -3.6 to -6.1 % for $r_{e3.7}$).

In reality, and especially for more heterogeneous clouds, the vertical penetration error will be combined with a number of other errors that affect both the $r_e$ and $\tau$, which are potentially larger and may compensate or enhance the bias due to vertical penetration depth. Therefore caution is required when applying the bias corrections; we suggest that they are only used for more homogeneous clouds.

# 1  Introduction

Clouds have a major impact on Earth's radiative balance (Hartmann et al., 1992) and small changes in their properties are predicted to have large radiative impacts (e.g., Latham et al., 2008). The amount of shortwave (SW) flux reflected by fully overcast warm (liquid water) clouds for a given sun and scattering angle, or the reflectance of a cloud, is primarily determined by the cloud optical depth ($\tau$), which in turn can often be characterized by the liquid water path (LWP, the vertical integral of liquid water content) and the cloud droplet number concentration ($N_d$). For a given cloud updraft, $N_d$ is determined by the number concentration and physico-chemical properties of aerosols. Thus, couching cloud reflectance in terms of $N_d$ links the cloud albedo to aerosol and microphysical effects via the Twomey (1974) effect making $N_d$ a very useful quantity to determine observationally. $N_d$ can also influence cloud macrophysical feedbacks via its control on rain formation (Albrecht, 1989; Stevens et al., 1998; Ackerman et al., 2004; Berner et al., 2013; Feingold et al., 2015) and stratocumulus cloud top entrainment (Ackerman et al., 2004; Bretherton et al., 2007; Hill et al., 2009).

Satellite observations of clouds and $N_d$ are immensely useful for studying clouds, cloud–aerosol interactions and for model evaluation since they afford large spatial and temporal coverage. A method to obtain $N_d$ from passive satellite observations (e.g., from the MODerate Imaging Spectroradiometer, MODIS; Salomonson et al., 1998) of $\tau$ and the cloud droplet effective radius ($r_e$) for stratiform liquid clouds has been previously demonstrated (Han et al., 1998; Brenguier et al., 2000; Nakajima et al., 2001; Szczodrak et al., 2001; Boers et al., 2006; Quaas et al., 2006; Bennartz, 2007; Grosvenor and Wood, 2014; Bennartz and Rausch, 2017) and is described further below. For more details see the Grosvenor et al. (2018) review paper on this technique, which also describes the known sources of error. In cloudy environments, aerosol optical depth cannot be retrieved from satellite making cloud property observations such as $N_d$ and the cloud droplet effective radius ($r_e$) the only useful indicator of the influence of aerosol on clouds. An advantage to using $N_d$ rather than $r_e$ to study cloud–aerosol interactions is that $r_e$ is also determined by the cloud water content and thus is a function of cloud macrophysical properties. $N_d$ on the other hand is only weakly controlled by cloud macrophysics allowing some separation of microphysical and macrophysical effects.

However, retrievals of $N_d$ from space are still somewhat experimental and there is a lack of comprehensive validation of the retrievals and the assumptions required. There is a need to characterize and quantify the associated errors; in this paper we focus on doing this for one source of $N_d$ error using a one–year $N_d$ data set for stratocumulus clouds from MODIS.

## 2   The adiabatic $N_d$ and LWP retrieval model, and the vertical penetration depth bias

$N_d$ and LWP are retrieved from passive satellite retrievals of $r_e$ and $\tau$ using an adiabatic cloud model that is described below. However, as shown in Platnick (2000) and Bennartz and Rausch (2017), for a retrieval free from other error sources (e.g. those due to cloud heterogeneity), the retrieved $r_e$ is representative of the $r_e$ value lower down in the cloud due to the vertical penetration of photons at the shortwave infra-red wavelengths used to retrieve $r_e$. In contrast, the retrieved $\tau$ is comprised of contributions from the extinction coefficient $\beta_{ext}(h)$, where h represents height from cloud base, throughout the whole cloud profile :-

$$\tau = \int_0^H \beta_{\text{ext}}(h)dh. \tag{1}$$

Here h=0 represents cloud base and h=H is cloud top.

$\beta_{ext}(h)$ is defined as :-

$$\beta_{\text{ext}}(h) = \pi \int_0^\infty Q_{\text{ext}}(r)r^2 n(r)dr, \tag{2}$$

where $r$ is the droplet radius and $n(r)$ is the droplet size number distribution within a cloud unit volume such that $N_d = \int_0^\infty n(r)dr$. $Q_{\text{ext}}(r)$ represents the ratio between the extinction and the geometric cross section of a given droplet and can be approximated by its asymptotic value of 2 (van de Hulst, 1957) since droplet radii are generally much larger than the wavelength of light concerned (typically 0.6 to 0.85 $\mu$m) such that the geometric optics limit is almost reached.

$r_e$ and liquid water content $LWC$ at a given height are respectively defined as:

$$r_e(h) = \frac{\int_0^\infty r^3 n(r)dr}{\int_0^\infty r^2 n(r)dr} \tag{3}$$

and

$$LWC(h) = \frac{4\pi\rho_w}{3} \int_0^\infty r^3 n(r)dr, \tag{4}$$

where $\rho_w$ is the density of liquid water. Combining Eqns. 3 and 4 and inserting into Eqn. 2 gives :-

$$\beta_{\text{ext}}(h) = \frac{3Q_{\text{ext}}}{4\rho_w} \frac{LWC(h)}{r_e(h)} \tag{5}$$

To determine the form of $r_e(h)$ in the above equation in terms of $L(h)$ and $N_d(h)$ we can utilize the fact that the "$k$" value,

$$k = \left(\frac{r_v}{r_e}\right)^3,$$
(6)

20   which is a measure of the width of the droplet size distribution (lower values indicate wider distributions), has been shown to be approximately constant in stratocumulus clouds (Martin et al., 1994; Pawlowska et al., 2006; Painemal and Zuidema, 2011). In this study we adopt a value of $k = 0.72$, which is the value assumed by the MODIS retrieval (Zhang, 2013). $r_v$ is the volume radius, defined as :-

$$r_v(h)^3 = \frac{1}{N_d(h)}\int_0^\infty r^3 n(r)dr = \frac{3LWC(h)}{4\pi\rho_w N_d(h)} = kr_e(h)^3,$$
(7)

where we have used Eqn. 4 to insert LWC and Eqn. 6 to write $r_v$ as a function of k and $r_e$. Now we utilize the assumptions that
$N_d$(h) is constant with height and that LWC(h) is a constant fraction, $f_{ad}$, of adiabatic. The latter equates to :-

$$LWC(h) = f_{ad}c_w h,$$
(8)

where $c_w$ is the rate of increase of LWC with height ($dLWC/dz$, with units $\mathrm{kgm^{-4}}$) for a moist adiabatic ascent and is referred to as the "condensation rate" in Brenguier et al. (2000), or the "water content lapse rate" in Painemal and Zuidema (2011). See Ahmad et al. (2013) for a derivation. $c_w$ is a constant for a given temperature and pressure. Allowing these assumptions, using
Eqn. 7 to substitute for $r_e$ in Eqn. 5 and combining with Eqns. 1 and 8 we can write :-

$$\tau^* = \int_0^{H^*} Q_{\text{ext}}\left(\frac{3f_{ad}c_w}{4\rho_w}\right)^{2/3}(N_d\pi k)^{1/3}h^{2/3}dh$$

$$= \frac{3Q_{\text{ext}}}{5}\left(\frac{3f_{ad}c_w}{4\rho_w}\right)^{2/3}(N_d\pi k)^{1/3}H^{*5/3}$$
(9)

At this stage, $H^*$ is any arbitrary height above cloud base and $\tau^*$ is thus the optical depth between the cloud base and that height. $H^*$ can be expressed as a function of $r_e(H^*)$, $k$, $N_d$ and some constants by using Eqns. 7 and 8. Then, given $r_e(H^*)$ and $\tau^*$, $N_d$ can be calculated as follows :-

$$N_d = \frac{\sqrt{5}}{2\pi k}\left(\frac{f_{ad}c_w\tau^*}{Q_{\text{ext}}\rho_w r_e(H^*)^5}\right)^{1/2}$$
(10)

Generally, when retrieving $N_d$ it is assumed that the $r_e$ obtained from satellite is representative of that from cloud top, i.e., $r_e(H^*)$=$r_e$(H) (e.g. Bennartz, 2007; Painemal and Zuidema, 2011). This would then mean that $\tau^*$ is the full cloud optical

depth ($\tau$) as retrieved by the satellite and thus could be used in Eqn. 10 above to obtain $N_d$. However, since the $r_e$ obtained by satellite is actually equal to $r_e(H^*)$ then $\tau^* < \tau$ and thus $\tau^*$ should be used in Eqn. 10 instead of the retrieved $\tau$; the problem lies in the fact that $\tau^*$ is unknown. However, in this paper we fit a simple function for $\tau^*$ as a function of $\tau$ based on radiative transfer modelling of a variety of idealised clouds.

Alternatively, Eqn. 10 can be formulated using the retrieved $\tau$ over the full cloud depth (setting $\tau^* = \tau$) and the cloud top $r_e$ (setting $r_e(H^*) = r_e(H)$). The problem then becomes one of estimating $r_e(H)$ from the retrieved $r_e(H^*)$. Here we formulate a parameterization of $r_e(H)$ / $r_e(H^*)$ as a function of $\tau$. Note, that either the $\tau$ or $r_e$ corrections should be applied to correct $N_d$, but not both together.

Then we estimate the error introduced in $N_d$ retrievals for one year of MODIS data due to the usual assumptions of $r_e(H^*){=}r_e(H)$ and $\tau^*{=}\tau$, on the assumption that there are no other biases affecting the $r_e$ retrieval. We label this bias the "vertical penetration bias".

The method of correcting $r_e$ has the advantage over the $\tau$ correction since it also allows a correction to the retrieval of LWP. LWP can be estimated (see e.g., Szczodrak et al., 2001) using :

$$\text{LWP} = \frac{5}{9}\rho_w r_e(H)\tau. \tag{11}$$

For a corrected LWP the cloud top $r_e$ and the retrieved (total) $\tau$ values should be used. Since the retrieved $r_e(H^*)$ is likely to be underestimated due to the vertical penetration depth bias, LWP would otherwise be underestimated and the correct value can be obtained by using the parameterized $r_e(H)$ instead.

## 3   Data and Methods

### 3.1   Calculation of $\tau$ and $r_e$ corrections

In order to calculate

$$g_{re} = \frac{r_e(H)}{r_e(H^*)} \tag{12}$$

and

$$d\tau = \tau - \tau^* \tag{13}$$

we have performed $r_e$ retrievals on idealised clouds using a similar algorithm to that used for MODIS retrievals. We produced idealised clouds that span a large range of stratocumlus–like clouds as represented by combinations of $N_d$ and LWP. We chose 41 values between $N_d{=}10$ and $1000\ \mathrm{cm}^{-3}$ that were equally spaced in log space and 91 values between LWP=20 and $200\ \mathrm{gm}^{-2}$ spaced equally in linear space. All of the possible combinations from this sampling were used to sample the 2-D ($N_d$, LWP)

phase space. For each combination, discretized adiabatic model profiles following the form of those described in Section 1 (i.e., with a vertically constant $N_d$ and LWC that increases linearly with height) were generated using $c_w$=1.81×10$^{-6}$ kg m$^{-4}$, $f_{ad}$=0.8 and using a vertical spacing of 1 m. The droplet size distributions at each height were represented by a modified gamma distribution with a $k$ value of 0.72, i.e. representative of an effective variance of 0.1. 1-D radiative transfer (RT) calculations, assuming plane-parallel clouds, were performed on these profiles using the DISORT (Discrete Ordinates Radiative Transfer

Program; Stamnes et al., 1988) radiation code in order to simulate reflectances at wavelengths of 0.86, 2.1 and 3.7 μm, matching those measured by MODIS to retrieve $\tau$ and $r_e$ over an ocean surface. Note that MODIS provides $r_e$ retrievals using both 2.1 μm and 3.7 μm wavelengths, which are hereafter referred to to as $r_{e2.1}$ and $r_{e3.7}$, respectively. The MODIS $r_{e3.7}$ retrieval requires a correction to account for the contribution to the observed radiance from thermal emission, which is based on the observed 11 μm radiance (Platnick and Valero, 1995; King et al., 2015; Platnick et al., 2017). We account for this in our retrievals by removing the thermal contribution during the RT calculation instead of via the 11 μm radiance, which should produce a consistent end result. The RT calculations were performed assuming a black surface, a clear atmosphere (i.e. gaseous absorption is neglected), using a solar zenith angle of 20° and a nadir viewing angle.

    These reflectances were then used to retrieve $\tau$ and $r_e$ values using the Nakajima and King (1990) bi-spectral method, as

operationally used by MODIS. To do so, a lookup table was built from reflectances calculated for a range of clouds that were assumed to be plane-parallel in nature, as assumed for the operational MODIS retrievals; i.e. these clouds were uniform in the vertical and horizontal with infinite horizontal extent. Again, a black surface and a $k$ value of 0.72 were assumed along with the same viewing geometry as for the RT calculations on the adiabatic clouds. A fixed depth of 1 km was assumed with cloud base at an altitude of 1 km and cloud top at 2 km, although the cloud depth has no major effect on the reflectances generated

for a given $\tau$ and $r_e$. $g_{re}$ was then calculated using the retrieved and model top $r_e$ values. $d\tau$ was calculated by choosing the value from the model profile of $\tau$, as measured from cloud top downwards, that corresponded to the location where the model profile $r_e$ matched the retrieved $r_e$.

    Figure 1a shows a 2-D histogram of $g_{re}$ values as a function of $\tau$ for the 2.1 μm retrieval. It shows that when plotted in this way $g_{re}$ forms a fairly tight relationship with $\tau$ so that for a given $\tau$ only a small range of $g_{re}$ values are possible. This

suggests that the relationship can be parameterized based upon a 1-D relationship fitted to this data with little loss of accuracy. The median value of each $\tau$ bin is also plotted (after smoothing over $\tau$ windows of 0.2) and this is the relationship used in this paper. $g_{re}$ is seen to decrease with $\tau$ with a gradient that decreases with $\tau$. Similarly, Figure 2 shows $d\tau$ vs $\tau$, which also shows a tight relationship that is suited to a 1-D parameterization. 4th order polynomial curves can be fitted (using the least squares method) to the median value relationships that take the form:-

$$g_{re} = a_4\tau^4 + a_3\tau^3 + a_2\tau^2 + a_1\tau + a_0 \qquad (14)$$

    and

$$d\tau = b_4\tau^4 + b_3\tau^3 + b_2\tau^2 + b_1\tau + b_0 \qquad (15)$$

The coefficients of these fits are given in Tables 1 and 2 along with the maximum errors for the fit (relative to the mean or median line) for the range shown. The curves (white lines in Figs. 1a and 2a) fit the mean data well with maximum absolute differences of 0.001 and 0.09, respectively, for the $g_{re}$ and $d\tau$ curves. However, there will be some error when using this relationship (or the mean value relationship) due to the spread in the $g_{re}$ and $d\tau$ values seen in the underlying histograms.

**Table 1.** Coefficients for the fitting curve (Eqn. 14) to estimate the median $g_{re}$ value as a function of $\tau$. The maximum absolute error between the fit and the median line is also shown.

| Retrieval wavelength | $a_4$ | $a_3$ | $a_2$ | $a_1$ | $a_0$ | Max abs $g_{re}$ error |
|---|---|---|---|---|---|---|
| 2.1 µm | 2.413e-07 | -2.467e-05 | 9.883e-04 | -0.02049 | 1.244 | 0.001 |
| 3.7 µm | 5.367e-07 | -5.179e-05 | 0.00186 | -0.03038 | 1.217 | 0.003 |

**Table 2.** Coefficients for the fitting curve (Eqn. 15) to estimate the mean $d\tau$ value as a function of $\tau$. The maximum absolute error between the fit and the mean line is also shown.

| Retrieval wavelength | $b_4$ | $b_3$ | $b_2$ | $b_1$ | $b_0$ | Max abs $\tau$ error |
|---|---|---|---|---|---|---|
| 2.1 µm | -3.174e-06 | 3.931e-04 | -0.021 | 0.5754 | 0.3216 | 0.09 |
| 3.7 µm | -1.281e-05 | 1.099e-03 | -0.03304 | 0.4168 | 0.6005 | 0.14 |

Figures 1b and 2b show the same results for the 3.7 µm retrieval. Again tight 1-D relationships are suggested. Here, though, $g_{re}$ and $d\tau$ values are lower for a given $\tau$ and the curves are steeper at lower $\tau$ values, but flatten off much more rapidly. By $\tau$=7.5 there is little dependence of $d\tau$ on $\tau$ and $d\tau$ saturates at a mean value of ∼2.6. The fit estimates for the curves (Eqns. 14 and 15 and Tables 1 and 2) again match the actual curves closely with a maximum absolute error in $g_{re}$ and $d\tau$ of 0.003 and 0.14, respectively.

## 3.2   MODIS data

For the MODIS data we use one year (2008) of MODIS Aqua data and follow a similar methodology to that used in Grosvenor and Wood (2014) in order to create a data set akin to the MODIS Level-3 (L3) product (King et al., 1997; Oreopoulos, 2005). We processed MODIS collection 5.1 joint Level-2 (L2) swaths into $1^o \times 1^o$ grid boxes. Joint-L2 swaths are sub-sampled versions of the full L2 swaths (sampling every 5th 1 km pixel) that also contains fewer parameters. We process the data from L2 to L3 in order to allow the filtering out of data at high solar zenith angles and to provide both $r_{e2.1}$ and $r_{e3.7}$ retrievals.

For this work we relax the screening methodology slightly from that used in Grosvenor and Wood (2014) since here we are interested in the effects of the vertical penetration $N_d$ bias upon a more general global data set. We applied the following

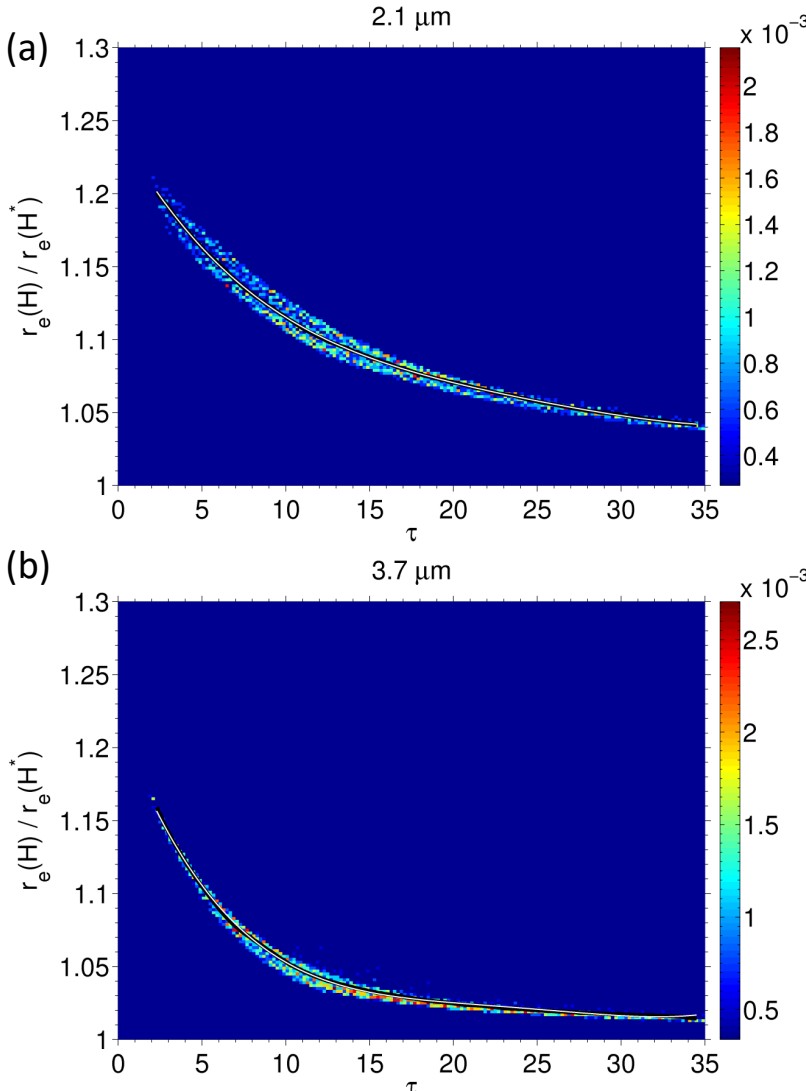

**Figure 1.** 2D histogram of $g_{re}$ as a function of $\tau$ for a range of clouds (see text) for the 2.1 μm $r_e$ retrieval (a) and the 3.7 μm retrieval (b). The black line is the median $g_{re}$ in each $\tau$ bin after smoothing over $\tau$ interval windows of 0.2. The white line is the fit to the mean curve using Eqn. 14.

restrictions to each $1^o \times 1^o$ sample that goes into the daily average (since multiple overpasses per day are possible) in order to attempt to remove some artifacts that may cause biases:

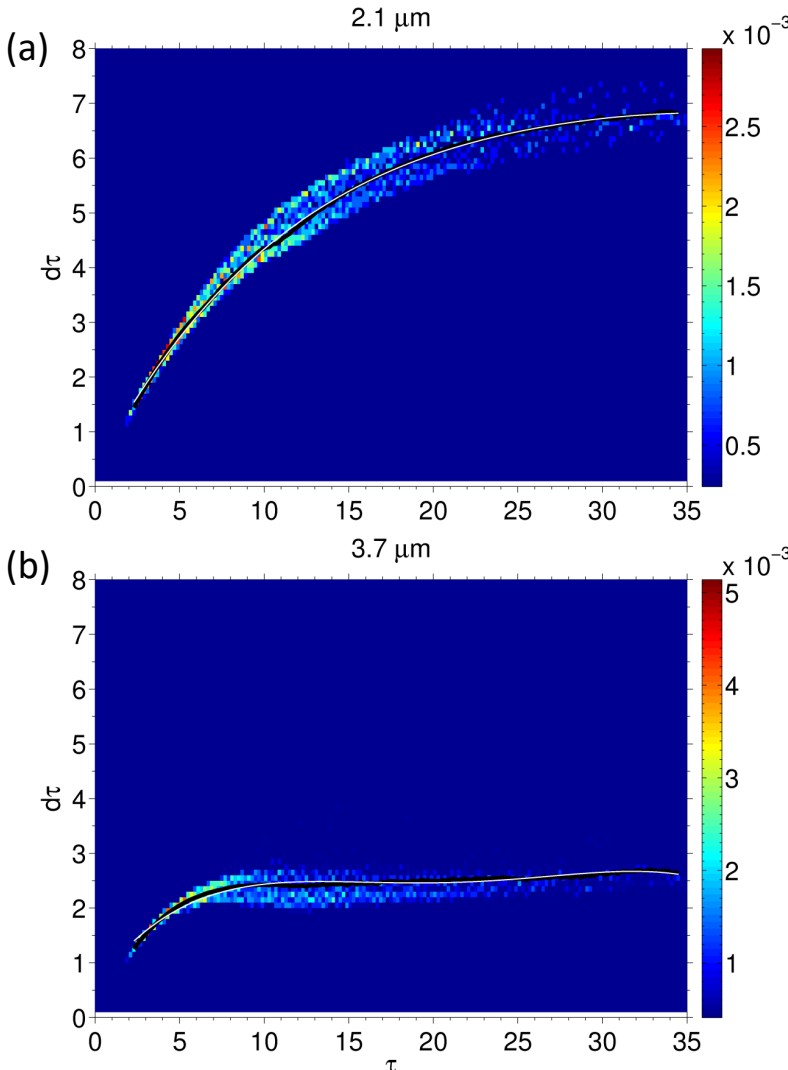

**Figure 2.** As for Fig. 1 except for $d\tau$ as a function of $\tau$ and using using Eqn. 15 for the white line.

1. At least 50 joint-L2 1 km resolution pixels from the MODIS swath that did not suffer from sunglint were required to have been sampled within each gridbox.

2. At least 80 % of the available (non-sunglint) pixels were required to be of liquid phase based upon the "primary cloud retrieval phase flag". Analysis was only performed on these pixels. A high cloud fraction helps to ensure that the clouds are not broken, since broken clouds are known to cause biases in retrieved optical properties due to photon scattering

through the sides of clouds. Often retrievals of $N_d$ are restricted to high cloud fraction fields for this reason (Bennartz, 2007; Painemal and Zuidema, 2011) and so we focus on such datapoints here.

3. Only the pixels remaining after (2) for which the "cloud mask status" indicated that the cloud mask could be determined, the "cloud mask cloudiness flag" was set to "confident cloudy", successful simultaneous retrievals of both $\tau$ and $r_e$ for the 2.1μm channel were performed and the cloud water path confidence from the MODIS L2 quality flags was designated as "very good confidence" (the highest level possible) were used. This is a little different from the official MODIS L3 product where a set of cloud products are provided that are weighted using the quality assurance flags. Rather than weighting our L3-like product with the QA flags we have simply restricted our analysis to pixels with the highest confidence for water path.

4. The mean $1^o \times 1^o$ cloud top height (CTH) is restricted to values lower than 3.2 km. This is done both to avoid deeper clouds for which $N_d$ retrievals are likely to be problematic due to the increased likelihood of a breakdown of the assumptions required to estimate $N_d$, such as a constant fraction of the adiabatic value for LWC and vertically constant $N_d$, as well increased retrieval issues due to cloud heterogeneity. CTH is calculated from the MODIS $1^o \times 1^o$ mean cloud top temperature (CTT) and the sea surface temperature (SST) using the method of Zuidema et al. (2009). SST data was obtained from the v2 of the NOAA Optimum Interpolation (OI) Sea Surface Temperature data set (NOAA_OI_SST_V2) that provides weekly SST data at $1^o \times 1^o$ resolution. This was interpolated to daily data on the assumption that SST does not vary significantly over sub-weekly timescales.

5. The mean $1^o \times 1^o$ solar zenith angle (SZA) was restricted to $\leq 65^o$ following the identification of biases in the retrieved $\tau$, $r_e$ and $N_d$ at high SZAs (Grosvenor and Wood, 2014).

6. $1^o \times 1^o$ grid-boxes were rejected if the maximum sea-ice areal coverage over a moving two week window exceeded 0.001 %. The sea-ice data used was the daily $1^o \times 1^o$ version of the "Sea Ice Concentrations from Nimbus-7 SMMR and DMSP SSM/I-SSMIS Passive Microwave Data, Version 1" data set (Cavalieri et al., 1996).

7. Only $1^o \times 1^o$ gridpoints with mean $\tau > 5$ were considered for the $N_d$ data set due to larger uncertainties from instrument error and other sources of reflectance error for $\tau$ and $r_e$ retrievals at low $\tau$ (Zhang and Plantnick, 2011; Sourdeval et al., 2016).

Following this screening, the $1^o \times 1^o$ gridboxes associated with each MODIS Aqua overpass were averaged into daily mean values for ocean covered surfaces only. Figure 3a shows the number of days from the year of data examined in this study (year 2008) that fulfilled the above criteria and thus are likely to produce a good $N_d$ retrieval. Regions with high numbers of days where useful $N_d$ retrievals can be made have been selected for closer examination in this study; they are listed in Table 3 along with information on the mean and maximum numbers of days of good data. The permanent marine stratocumulus decks are among those selected, namely those: in the SE Pacific off the western coast of S. America (Region #1); in the SE Atlantic off the western coast of southern Africa (Region #2); off the coast of California and the Baja Peninsula (Region #3); in the Bering

Sea off the SW coast of Alaska (Region #6); and in the Barents Sea to the north of Scandinavia (Region #8). These regions

are where the highest numbers of selected days occur with values ranging up to a maximum of 141 days (for the Bering Sea

region). The Barents Sea region has the lowest maximum number of days out of this group, reflecting the fact that $N_d$ retrievals

cannot be made during a lot of the winter season in this region due to a lack of sunlight. The Southern Ocean (Region #5) and

the NW Atlantic (Region #7) regions frequently produce stratocumulus, although it is often associated with the cold sectors of

cyclones and so its location from day to day is more transient. These regions are also affected by high solar zenith angles in the

winter seasons, which also restricts the number of retrievals possible there. The East China Sea region (Region #4) produces

the lowest mean and maximum numbers of days since the stratocumulus areas are mostly restricted to near the coast and occur

mostly in the winter season.

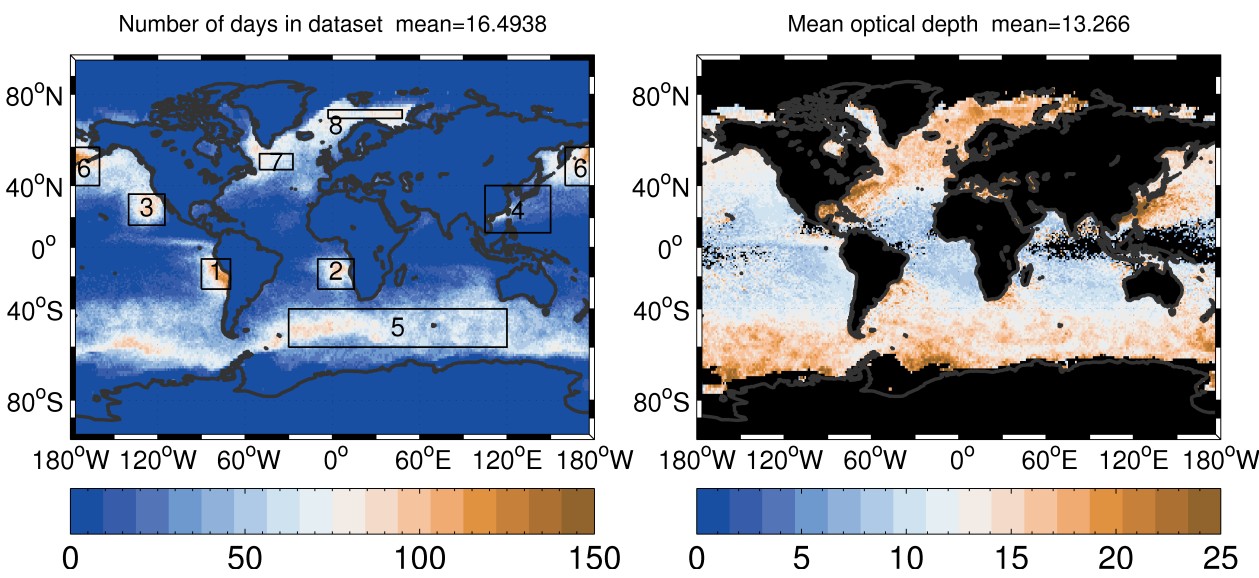

**Figure 3.** Left: Number of days in 2008 that fulfilled the criteria required to be counted as a valid $N_d$ retrieval. See the text for details on the criteria. Various regions of interest are also denoted by the boxes and numbers. Right: Mean optical depth for data set with filtering criteria 1-7 applied (see text).

$N_d$ was calculated for both $r_{e2.1}$ and $r_{e3.7}$ using Eqn. 10 from the $1^o \times 1^o$ daily mean $\tau$, $r_e$ and CTT. This was done by using

5   the retrieved $\tau$ value in Eqn. 10 along with both the retrieved $r_e$ value (i.e., assuming that $r_e(H^*){=}r_e(H)$ as is often assumed

for $N_d$ retrievals) and by estimating $r_e(H^*)$ using the retrieved $r_e$ along with the $g_{re}$ values that were calculated as described

above. This therefore gives $N_d$ data sets for the "standard" method and a corrected method, allowing the differences between

the two to be examined. A similar process was applied for the LWP retrieval.

## 4   Results

10  Following Eqn 10, the ratio between the uncorrected and corrected $N_d$ values can be shown to be :-

$$\frac{N_{d(uncorrected)}}{N_{d(corrected)}} = \left(\frac{r_e(H)}{r_e(H^*)}\right)^{5/2} = g_{re}^{5/2} \tag{16}$$

Figure 4 shows how the relative $N_d$ bias varies as a function of retrieved $\tau$ when using an $r_e$ that has been corrected using the $g_{re}$ from Fig. 1 (mean curve, black line). At $\tau$=5 the relative error is 46 % for the $r_{e2.1}$ retrieval and 28 % for the $r_{e3.7}$ retrieval. At higher $\tau$ the errors reduce rapidly, but remain above 10% for the $r_{e2.1}$ retrieval over the $\tau$ range shown. For the $r_{e3.7}$ retrieval the relative error drops below 10% for $\tau >\sim 13$. Thus, the overall degree of error due to this effect will be determined by the

5  distribution of $\tau$ for the regions of interest, which we take into consideration here using MODIS data for a representative $N_d$ data set.

Alternatively, if the correction is formulated in terms of a correction to $\tau$ we obtain :-

$$\frac{N_{d(uncorrected)}}{N_{d(corrected)}} = \left(\frac{\tau}{\tau^*}\right)^{1/2} = \left(\frac{\tau}{\tau - d\tau}\right)^{1/2} \tag{17}$$

The equation shows that, for a constant $d\tau$, the relative $N_d$ bias due to an uncorrected $\tau$ value would increase with decreasing

10  $\tau$ as $\tau$-$d\tau$ approaches zero.

Figure 3b shows the time–mean $\tau$ for the data set as filtered by criteria 1-7 above; i.e. to replicate the type of filtering that would likely be performed for $N_d$ retrievals. Table 3 lists the regional means of these time–mean values along with the regional means of the standard deviations of $\tau$ over time. It shows that the mean $\tau$ values of the tropical and sub-tropical regions are generally lower than those at higher latitudes. The East China Sea, Barents Sea, NW Atlantic and Southern Ocean regions

exhibit the highest mean $\tau$ values out of those examined and so should be expected to show the lowest $N_d$ biases due to the vertical penetration effect. The SE Atlantic region (and the region to the west of Africa in general) show low $\tau$ and can be expected to give high $N_d$ biases. Table 3 also lists the fraction of days for which $\tau \leq 10$ ($f_{\tau \leq 10}$). $\tau$=10 is the value above which $N_d$ biases drop below 31% for the 2.1 μm retrieval and below 14 % for $r_{e3.7}$ according to Fig. 4. Thus $f_{\tau \leq 10}$ indicates the fraction of days for which daily $N_d$ biases will be greater than 31% for that channel. The values in the table indicate that

even in the least affected region (Barents Sea) this will occur for 21% of the days. For the SE Atlantic and SE Pacific region the percentages rise to 69% and 53% of the days, respectively. Thus, the vertical penetration depth $N_d$ bias is prevalent in all regions for which $N_d$ data sets are likely to be used, and particularly so in the sub–tropical stratocumulus regions where $N_d$ retrievals have been widely used and studied.

The overall bias is now estimated using one year of actual MODIS data in order to obtain a realistic distribution of $\tau$ values.

However, it should be noted that the data set used is deliberately filtered in order to only retain datapoints that are likely to give useful $N_d$ data, namely low liquid clouds with extensive 1×1º cloud fractions; i.e., predominately stratocumulus. This is done in order to assess biases for the types of clouds that $N_d$ data sets will typically be used to study.

**Table 3.** Regional statistics for the various marine stratocumulus regions shown in Fig. 3. Shown are the mean and maximum number of days that fulfill the screening criteria in order to be considered as useful $N_d$ retrievals; the regional means and standard deviations ($\sigma$) of the time-averaged optical depths ($\tau$) for the screened data set; and the regional mean of the fraction of days for which $\tau \leq 10$ ($f_{\tau \leq 10}$), which is calculated using only data from gridpoints for which the number of days with $N_d$ data was $\geq 15$.

| # | Region name | Mean no. days | Max no. days | Mean $\tau$ | $\sigma_\tau$ | $f_{\tau \leq 10}$ |
|---|-------------|---------------|--------------|-------------|---------------|---------------------|
| 1 | SE Pacific | 68.3 | 132 | 10.5 | 3.81 | 0.53 |
| 2 | SE Atlantic | 52.6 | 107 | 9.1 | 3.12 | 0.69 |
| 3 | California | 62.4 | 114 | 10.5 | 4.06 | 0.54 |
| 4 | East China Sea | 12.9 | 77 | 18.3 | 10.13 | 0.24 |
| 5 | Southern Ocean | 58.2 | 101 | 14.2 | 7.58 | 0.35 |
| 6 | Bering Sea | 73.4 | 141 | 13.6 | 6.84 | 0.36 |
| 7 | NW Atlantic | 64.3 | 90 | 15.9 | 9.22 | 0.29 |
| 8 | Barents Sea | 74.9 | 88 | 18.0 | 9.87 | 0.21 |

**Table 4.** Regional means of the predicted time-mean percentage biases in $N_d$ and LWP due to the vertical penetration depth error; and regional means of the relative (percentage) standard deviations (over time) of the percentage $N_d$ and LWP biases (i.e., regional means of the values in Fig. 6 and the equivalent for LWP). Bias results are shown for both the 2.1 μm and the 3.7 μm $r_e$ retrievals.

| # | Region name | 2.1 μm % $N_d$ biases Mean bias (%) | $\sigma$ (%) | 3.7 μm % $N_d$ biases Mean bias (%) | $\sigma$ (%) | 2.1 μm % LWP biases Mean bias (%) | $\sigma$ (%) | 3.7 μm % LWP biases Mean bias (%) | $\sigma$ (%) |
|---|-------------|------|------|------|------|------|------|------|------|
| 1 | SE Pacific | 31.9 | 21.7 | 15.0 | 37.0 | -10.4 | 18.2 | -5.4 | 33.6 |
| 2 | SE Atlantic | 34.5 | 18.6 | 17.1 | 32.0 | -11.1 | 15.5 | -6.1 | 29.0 |
| 3 | California | 32.0 | 22.3 | 15.1 | 37.5 | -10.4 | 18.8 | -5.4 | 34.2 |
| 4 | East China Sea | 24.6 | 36.1 | 10.7 | 53.5 | -8.3 | 31.3 | -3.9 | 49.5 |
| 5 | Southern Ocean | 27.5 | 31.6 | 12.1 | 49.0 | -9.1 | 27.2 | -4.4 | 45.1 |
| 6 | Bering Sea | 28.0 | 29.5 | 12.4 | 46.6 | -9.3 | 25.3 | -4.5 | 42.8 |
| 7 | NW Atlantic | 25.9 | 34.3 | 11.2 | 52.1 | -8.7 | 29.7 | -4.1 | 48.0 |
| 8 | Barents Sea | 23.7 | 37.4 | 9.9 | 56.0 | -8.0 | 32.4 | -3.6 | 51.6 |

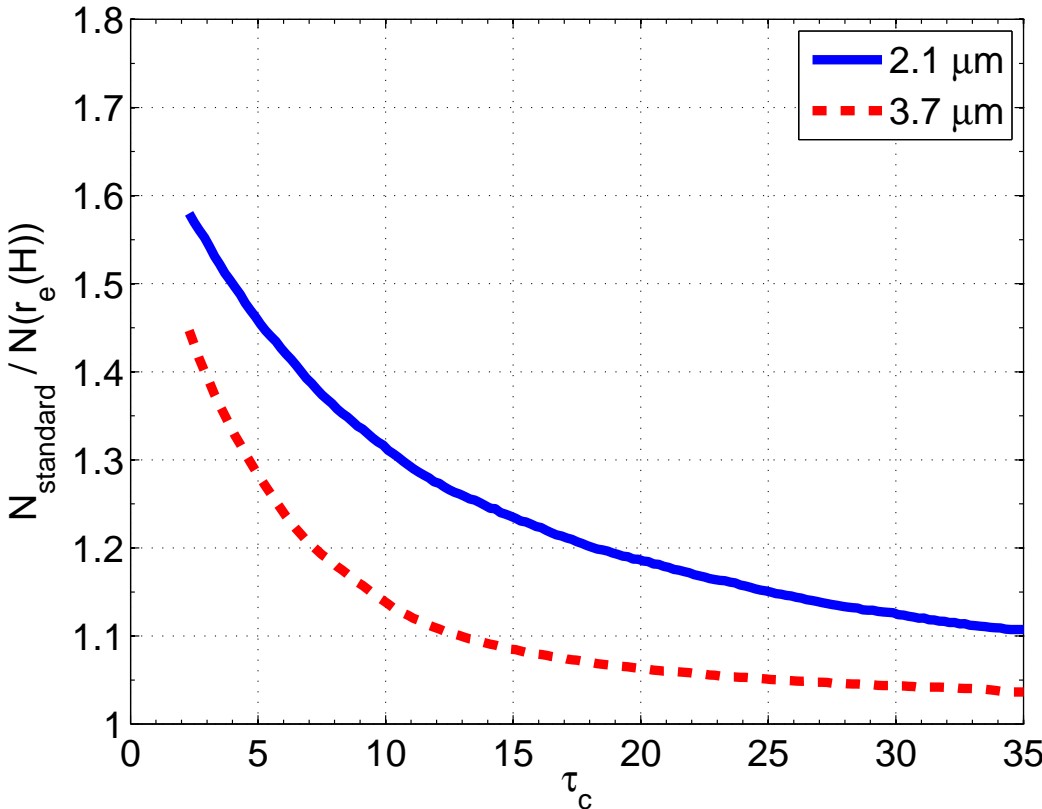

**Figure 4.** The ratios of $N_d$ values from the standard MODIS calculation (using the retrieved $r_e$ for $r_e^*$ in Eqn. 10) to those from the corrected calculation (using the corrected $r_e$ for $r_e^*$ as calculated from the retrieved value and $g_{re}$; the $g_{re}$ values used are those shown by the black line in Fig. 1) vs retrieved $\tau$.

Fig. 5 shows a map of the mean percentage biases and Table 4 gives the regional means of the values in the map. Considering firstly the biases for the $r_{e2.1}$ retrieval, the biases are highest in the tropics and sub-tropics. The regional mean bias is 34.5 % for the SE Atlantic region (Region #2), which is the stratocumulus region that seems to suffer the most. The biases are a little lower for the other major stratocumulus regions; e.g. for the SE Pacific region (Region #1) and the Californian region (Region #3) the mean biases are 32%, although the biases increase further west where the dominant cloud regime tends to shift towards trade cumulus clouds. The remaining regions all have mean biases of 24–28%. The Barents Sea region (Region #8) has a value of only 23.7%, representing the stratocumulus region with lowest mean bias. These results indicate higher $\tau$ values for the clouds in the East China Sea, Southern Ocean, Bering Sea, NW Atlantic and Barents Sea regions relative to the Californian and S.E. Pacific stratocumulus regions, with the SE Atlantic region exhibiting the lowest $\tau$ values. This is confirmed by the mean $\tau$ values shown in Table 3. The biases for the $r_{e3.7}$ retrieval display the same spatial patterns as for $r_{e2.1}$, but are significantly

lower; the mean value in the region with the maximum bias (SE Atlantic, Region #2) is 17% and that in the region with the
lowest bias (Barents Sea, Region #8) is 10%.

The regional mean LWP biases are also listed in Table 4. They are negative since an $r_e$ underestimate from the vertical
penetration effect leads to an LWP underestimate (see Eqn. 11). The biases are also smaller in magnitude than for $N_d$ due to
the smaller sensitivity of LWP to $r_e$ inherent in the latter equation. They are anti-correlated with the $N_d$ biases such that the
region with largest $N_d$ bias (SE Atlantic) has the largest negative LWP bias of -11.1 %. The smallest magnitude bias occurs in
that Barents Sea region (-8 %).

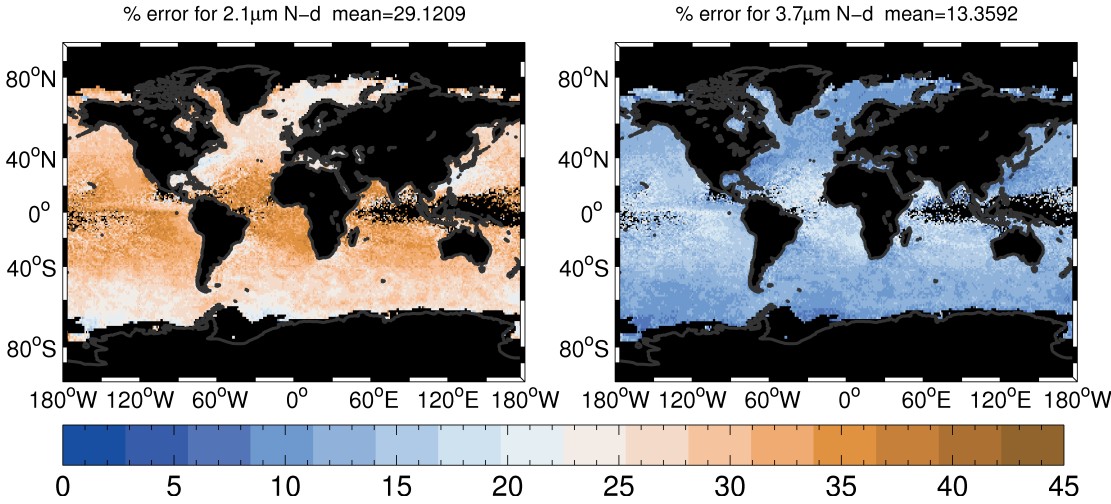

**Figure 5.** Maps of the annual mean percentage error for uncorrected $N_d$ retrievals using a year (2008) of daily MODIS data that has been
filtered to select data points in which $N_d$ retrievals are favourable and therefore most likely to be used for $N_d$ data sets (see text for details).
The left plot shows the results for the $r_{e2.1}$ retrieval and the right for the $r_{e3.7}$ one.

It is also useful to know how variable the biases are from day to day for a given point in space since this will determine how
useful the application of a single offset bias correction might be for correcting $N_d$ biases for daily data. Figure 6 shows the
time variability of the bias in the form of the relative standard deviations (over time) of the percentage $N_d$ biases. It reveals that
the percentage bias in $N_d$ generally has a larger relative standard deviation at latitudes above around 40$^o$ with values typically
ranging up to around 30–50% (of the mean percentage $N_d$ bias) for the 2.1 μm retrieval. Relative variability is greater for the
3.7 μm retrieval, perhaps due to the much lower mean percentage errors. Some of the selected regions show more variability
than others, in particular the Barents Sea and East China Sea regions.

Table 4 gives the regional means of the relative standard deviations revealing values that range from approximately 20 to
40% of the mean percentage biases for the 2.1 μm retrieval and 30–60% for the 3.7 μm one. This shows that the application
of a single annual mean offset bias correction is likely to lead to fairly large biases for the $N_d$ estimates for individual days for
regions where the mean $N_d$ errors are significant. If daily data is used to determine relationships between cloud properties and

5   $N_d$ without correcting for the biases examined here then significant variability in $N_d$ might be introduced that may affect those

relationships via non-linear effects.

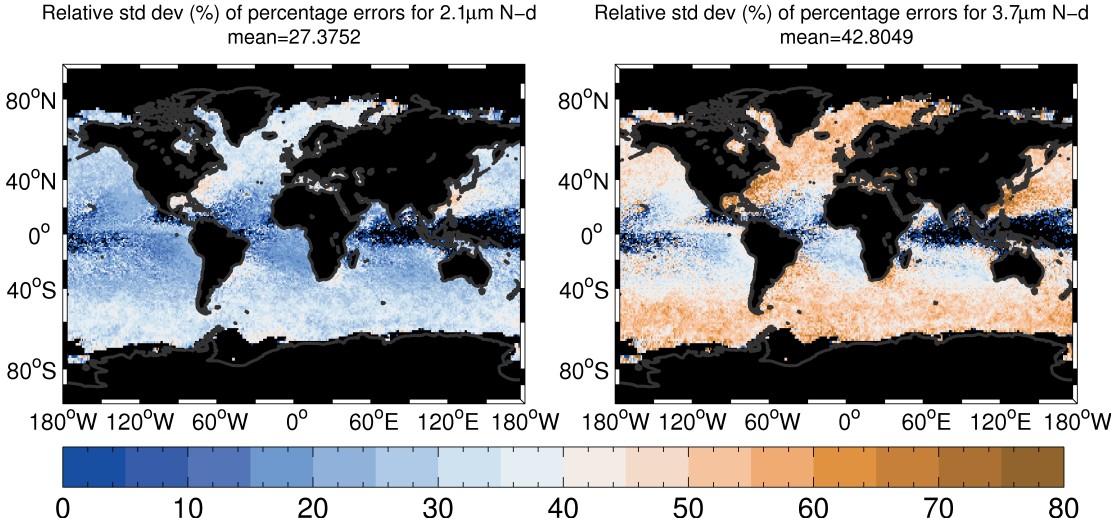

**Figure 6.** As for Fig. 5 except showing the relative (as a percentage) standard deviation of the percentage $N_d$ bias over time, i.e. $\sigma_{\%bias}/\overline{\%bias}$.

    Fig. 7 shows how the percentage $N_d$ biases change with season for the $r_{e2.1}$ retrieval only. Interestingly, the highest biases tend to occur in the DJF season for the SE Pacific and SE Atlantic stratocumulus regions, indicating that $\tau$ values are lower in DJF for those seasons. The SON season also generally produces higher biases than MAM and JJA for those regions, particularly for SE Pacific. For the East China Sea region the biases are lower in SON and DJF seasons than in the other seasons. We note that there is little data in this region for JJA since there are few low-altitude clouds with large regional liquid cloud fractions there in this season. The other regions either do not show a large amount of seasonal variability, or $N_d$ data is only available for part of the year due to a lack of sunlight in the winter months.

**5  Discussion**

There are some caveats to the results that we presented here that we now discuss. We have shown that, theoretically, the effect of retrieving a lower $r_e$ than the cloud top $r_e$ can be corrected for by simply replacing $r_e$ with the parameterized cloud top version, or by removing $d\tau$ from the observed $\tau$. However, this rests upon the parameterizations being valid across all of the cloud types relevant for the $N_d$ and LWP data sets. The relationships are based on the retrieved $r_e$ for a range of clouds,

although only for a nadir viewing angle and a solar zenith angle of $20^o$. Platnick (2000) showed that $d\tau$ has some dependence on viewing geometry and so the consideration of a wider range of view and solar zenith angles should ideally be made.

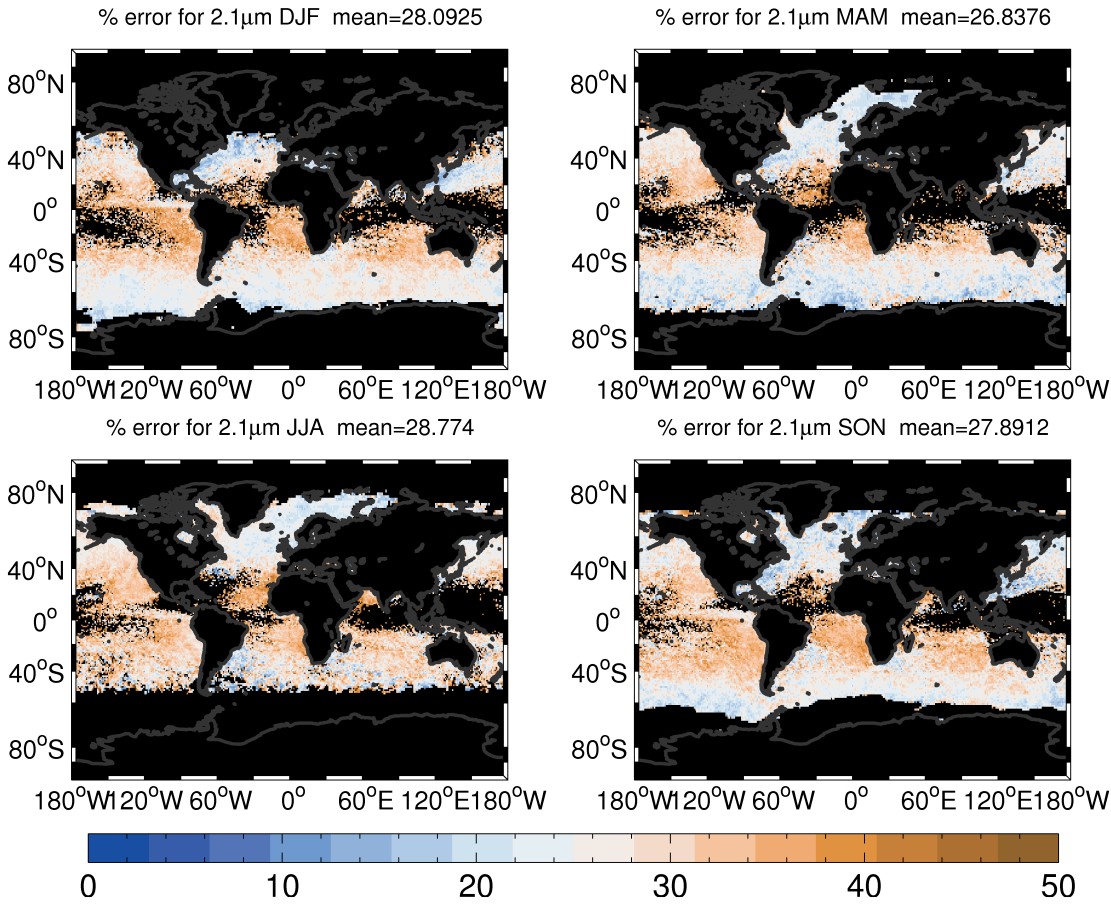

**Figure 7.** Seasonal mean percentage $N_d$ biases for the $r_{e2.1}$ retrieval only.

In addition, a liquid water condensation rate ($c_w$) value of $1.81 \times 10^{-6}\,\mathrm{kg\,m^{-4}}$ was assumed for the model adiabatic clouds, which corresponds to a cloud temperature of 278 K at a pressure of 850 hPa. In reality, cloud temperatures and hence $c_w$ will vary, mainly as a function of cloud temperature. We have performed sensitivity tests using a value of $1.0 \times 10^{-6}\,\mathrm{kg\,m^{-4}}$,

which corresponds to a cloud temperature of 262 K. This is likely to be close to the coldest temperature attained by boundary layer clouds over the oceans that are coupled to the surface, which are generally the types of clouds for which the droplet concentration retrievals are applied. The results (not shown) reveal mean differences (across the $\tau$ values tested) in the mean $d\tau$ line (i.e., the white line in Fig. 2a) of 4.2 % (the maximum difference was 10 %) for the 2.1 μm retrieval and 3.6 % (maximum of 7.6 %) for the 3.7 μm one. For $g_{re}$ the differences were of the opposite sign and much smaller, with mean

differences in the median $d\tau$ line of -0.45 % (the most negative difference was -0.6 %) for the 2.1 μm retrieval and -0.3 % (-0.66 % was the most negative difference) for the 3.7 μm one. Therefore, the effect of $c_w$ changes is relatively minor. These results would also apply for equivalent changes in the cloud adiabaticity (i.e., the value of $f_{ad}$).

The modelling of the idealised clouds and the correction rests on the assumption that $r_e$ increases monotonically with height within the cloud (following the adiabatic assumption), but there is some suggestion that the development of precipitation-sized

droplets might lead to larger droplets being preferentially found below cloud top (Chang and Li, 2002; Nakajima et al., 2010a, b; Suzuki et al., 2010) . However, Zhang et al. (2012) found that MODIS retrievals of $r_e$ performed on model generated clouds were not significantly affected by the presence of precipitation. Also, during the VOCALS field campaign in the SE Pacific region, aircraft observations showed that $r_e$ generally did increase with height up to cloud top (Painemal and Zuidema, 2011), indicating that this is not a problem at least for the near-coastal clouds tested. Further offshore the likelihood of precipitation

increases as clouds become more cumulus-like and so for those clouds the issue may be greater and hence more caution should be exercised when interpreting the results presented here for such regions.

Evaporation effects related to entrainment also have the potential to reduce $r_e$, $N_d$ and $LWC$ near cloud top and hence negate some of the assumptions upon which the $N_d$ retrievals rest. However, we argue that the entrainment effect upon $r_e$ is likely to be minimal for two reasons: Firstly, the evidence suggests that for stratocumulus clouds extreme inhomogeneous mixing occurs

at cloud top, which reduces the $LWC$ and $N_d$, but does not change $r_e$ (Burnet and Brenguier, 2007; Brenguier et al., 2011; Painemal and Zuidema, 2011). Secondly, the results of Painemal and Zuidema (2011) indicate that entrainment occurs within approximately the first 0.5 optical depths from cloud top on average; the penetration depths calculated here are considerably larger than this for reasonably thick clouds (Fig. 2). The effect of the reduced $N_d$ and $LWC$ within the entrainment zone is not so clear-cut; this would negate the assumption of a vertically constant $N_d$ and monotonically increasing $LWC$ used to

formulate the total $\tau$. However, given the likely small $\tau$ contribution from the entrainment region relative to the total $\tau$, this effect is likely to be small.

It is also clear that the suggested correction for the vertical penetration effect should only be applied to the retrievals of $N_d$ with consideration of other bias sources. These other potential error sources are numerous and include $r_e$ biases due to sub-pixel heterogeneity (Zhang and Plantnick, 2011; Zhang et al., 2012, 2016); 3D radiative effects (Marshak et al., 2006);

assumptions regarding the degree of cloud adiabaticity ($f_{ad}$ in Eqn. 10; Janssen et al., 2011; Merk et al., 2016); the choice of $k$ value (assumed constant; Brenguier et al., 2011; Merk et al., 2016); the assumption of a vertically uniform $N_d$; the assumed droplet size distribution shape and width (Zhang, 2013); viewing geometry effects (Várnai and Davies, 1999; Horváth, 2004; Varnai and Marshak, 2007; Kato and Marshak, 2009; Liang et al., 2009; Di Girolamo et al., 2010; Maddux et al., 2010; Liang and Girolamo, 2013; Grosvenor and Wood, 2014; Liang et al., 2015; Bennartz and Rausch, 2017); upper level cloud and aerosol

layers (Haywood et al., 2004; Bennartz and Harshvardhan, 2007; Davis et al., 2009; Meyer et al., 2013; Adebiyi et al., 2015; Sourdeval et al., 2013, 2016), etc. These errors have the potential to bias $N_d$ in a way that opposes the positive bias expected from the vertical penetration effect such that the overall biases may cancel out. Indeed, the largest source of error in $N_d$ is likely that from $r_e$ biases given the sensitivity of $N_d$ to $r_e$ in Eqn. 10. MODIS $r_e$ has generally been shown to be biased positively compared to aircraft observations (Painemal and Zuidema, 2011; King et al., 2013), which would lead to a negative $N_d$ error

when taken alone. Thus, the application of the correction described in this paper in isolation has the potential to enhance any negative bias in $N_d$ caused by a positive $r_e$ bias.

Our paper quantifies the vertical penetration bias in isolation to the other effects mentioned above. It should be questioned, though, whether the presence of cloud heterogeneity and other effects somehow prevent the effects of the vertical stratification from influencing the retrieved $r_e$, making it irrelevant. This could be a potential explanation for why it is often observed that $r_{e2.1}$ is larger than $r_{e3.7}$ (Zhang and Plantnick, 2011) in contrast to the direction expected from adiabatic clouds given the vertical penetration effect, since it is known that sub-pixel heterogeneity effects tend to cause a positive $r_{e2.1}$ bias relative to $r_{e3.7}$ (Zhang et al., 2012). We argue, though, that the vertical stratification effect occurs in addition to other effects (heterogeneity, etc.) with the latter cancelling out and often exceeding the former such that the positive difference between $r_{e2.1}$ and $r_{e3.7}$ would be even larger without the vertical stratfication effect. The cancellation of biases may also explain why VOCALS aircraft measurements (Painemal and Zuidema, 2011) tended to show that $r_{e2.1}$ and $r_{e3.7}$ were very similar.

We also note that there are many situations when the expected result due to vertical stratification of $r_e$ does occur (i.e. $r_{e3.7} > r_{e2.1}$), as demonstrated in Painemal et al. (2013) and Fig. 8. This shows ratios between $r_{e3.7}$ and $r_{e2.1}$ for an example MODIS scene in the SE Pacific stratocumulus region. Ratios using the uncorrected MODIS $r_e$ values are shown, which shows that the ratio exceeds one for most of the stratocumulus cloud region (the clouds that adjoin the coast) with ratios ranging from around 1.1 to 1.2. In the more broken clouds the ratio is less than one, which is likely a result of cloud heterogeneity. However, it would be expected that $N_d$ retrievals would not be applied to such clouds. The figure also shows the ratios calculated using $r_{e3.7}$ and $r_{e2.1}$ values that have been corrected using the $g_{re}$ factors. If the differences between $r_{e3.7}$ and $r_{e2.1}$ were caused by vertical stratification alone and if our parameterization was correctly predicting the cloud top $r_e$ for both MODIS channnles then this ratio should be equal to one. This is the case for the clouds close to the coast indicating that our parameterization is working well for these clouds. The ratios are a little higher than one further north and west (around 1.05-1.08) indicating that either our parameterization is not working correctly for these clouds, or that other factors are causing relative differences between $r_{e3.7}$ and $r_{e2.1}$ (e.g. sub-adiabaticity, cloud heterogeneity, etc.). Figure 9 shows the percentage of pixels for which $r_{e3.7} > r_{e2.1}$ for 90 days of MODIS SE Pacific observations divided into four different heterogeneity bins. Heterogeneity is characterized by the $H_\sigma$ parameter (Liang et al., 2009), which is the standard deviation of the 250 m resolution 0.86 µm reflectance ($R_{0.86}$) divided by the mean $R_{0.86}$. It is clear that for many regions relative $r_e$ values that are consistent with an adiabatic profile occur more than 50% of the time, particularly when the cloud heterogeneity is low. This suggests that it may be possible to use $H_\sigma$ to determine the situations in which the bias correction is more applicable. However, it is hard to definitively prove our argument within the scope of this study, particularly for more heterogeneous regions, since it would likely require computationally expensive 3D radiative transfer modelling of known cloud fields (e.g., from LES models), followed by $r_e$ and $\tau$ retrievals.

Painemal and Zuidema (2011) actually demonstrated that MODIS $N_d$ agreed rather well with $N_d$ from aircraft for the SE Pacific region despite a fairly large positive $r_e$ bias; this was thought to be due to the fortuitous cancellation of (for $N_d$) the $r_e$ bias with biases in the $k$ parameter and $f_{ad}$. However, the agreement between aircraft and MODIS $N_d$ seen in Painemal and Zuidema (2011) would deteriorate if a correction for the $N_d$ bias due to the penetration depth effect discussed here was also applied. Table 4 indicates that the result would be a MODIS $N_d$ underestimate of around 32 % (average for SE Pacific,

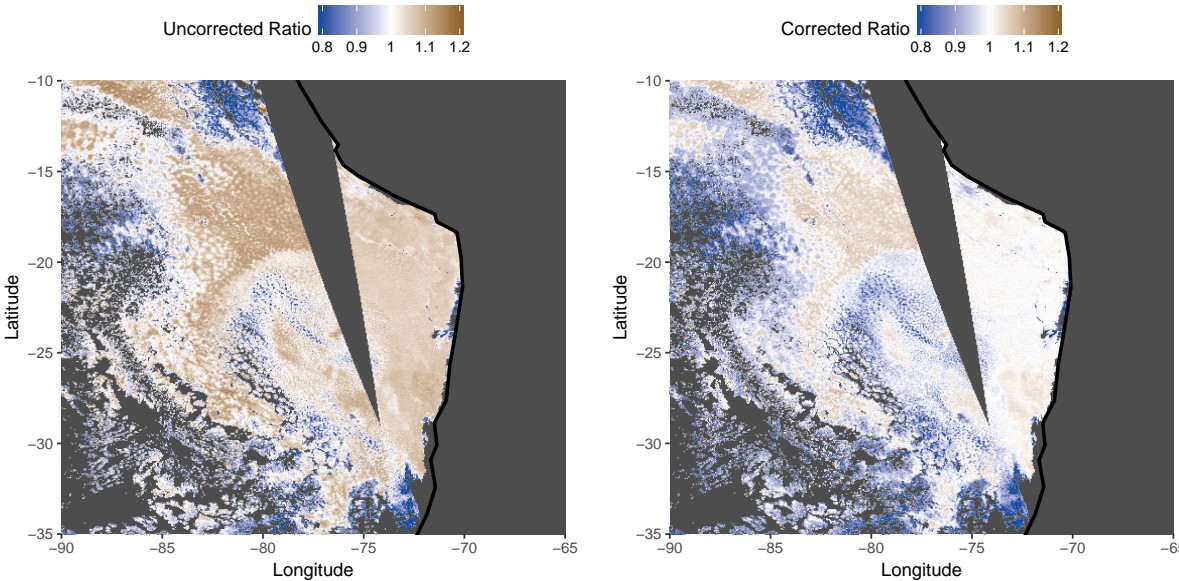

**Figure 8.** Ratios of $r_{e3.7}$ to $r_{e2.1}$ for a MODIS snapshot scene from the Southeast Pacific stratocumulus region from 16th June, 2015. Left: using uncorrected $r_e$ values. Right: using $r_e$ values that have been corrected using the parameterizations for $g_{re}$ (for both $r_{e3.7}$ and $r_{e2.1}$). A ratio of one is expected for the plot on the right if the relative differences between $r_{e3.7}$ and $r_{e2.1}$ are caused by vertical stratification alone and if the parameterization is correctly predicting the relative differences.

region#1) for the 2.1 μm retrieval, assuming perfect initial agreement. This indicates that another $N_d$ bias may have been operating in order to give the good observed agreement.

The MODIS retrieval uses reflectances from both a visible and a shortwave infra-red (SWIR) wavelength channel with the former being primarily determined by $\tau$ and the latter by $r_e$. However, a bi-spectral retrieval is used and so there is also some sensitivity of the retrieved $\tau$ to the SWIR reflectance, which will be representative of the $r_e$ below cloud top due to the vertical penetration effect. This, combined with the fact that the MODIS forward retrieval model assumes a vertically uniform cloud, will result in the retrieved $\tau$ being biased relative to the real value (assuming the real cloud has an adiabatic profile). Figure 10 shows the difference between the retrieved and model profile $\tau$; the bias is negative and smaller in magnitude than 5 % for the 3.7 μm retrieval. They are slightly larger for the 2.1 μm retrieval, but still lower in magnitude than 5 %, except at $r_e \lesssim 7$ μm. Although it should be noted that some of this bias may be due to other causes related to the inconsistencies between the vertically uniform and adiabatic models, rather than the $r_e$ vertical penetration bias. Since the retrieved $N_d$ is proportional to the square root of $\tau$, this will lead to small $N_d$ biases. Biases in LWP will be similar to those in $\tau$ since LWP is proportional to $\tau$, but $r_e$ biases are still likely to dominate (e.g., see Fig. 1). Thus, we have not pursued this further.

In this paper we have only considered retrievals over the ocean, although retrievals over land for $\tau$ and $r_e$ are available for MODIS. MODIS surface albedo uncertainties are likely to be much higher over land than over the oceans (King et al., 2004; Rosenfeld et al., 2004; Bréon and Doutriaux-Boucher, 2005) since the surface albedo is much more variable over land. In

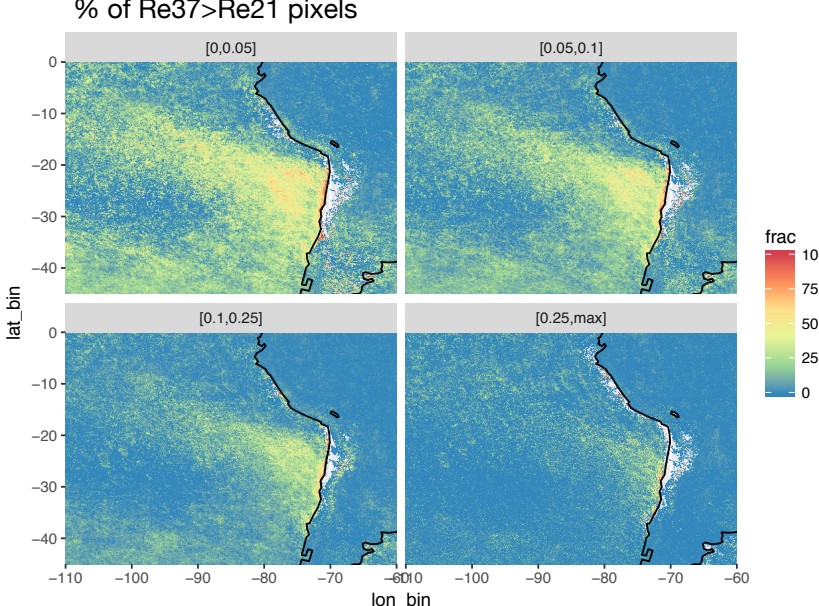

**Figure 9.** The percentage of pixels for which $r_{e3.7} > r_{e2.1}$ for 90 days (January, February and March of 2008) of $0.1^o$ resolution MODIS Collection 6 observations for the SE Pacific stratocumulus region. Only single layer liquid clouds are included and datapoints have been filtered to exclude $\tau < 5$ and partially cloudy pixels. The four panels are for four different bins of the heterogeneity parameter (the standard deviation of the 250 m resolution 0.86 μm reflectance divided by the mean reflectance) with bin ranges labelled above the panels.

addition, cloud masking is more difficult over land, particularly over non–vegetated surfaces, transitional areas between desert and vegetated surfaces and above high-altitude regions (Platnick et al., 2003). We have ignored land regions in order to avoid

5 such complications and also because stratocumulus clouds are more prevalent over ocean regions (Klein and Hartmann, 1993; Wood, 2012). However, the results shown in this paper may still apply over land. The results of Rosenfeld et al. (2004) and Platnick et al. (2017, their Fig. 14) suggest that surface albedo uncertainties are more important at lower optical depths ($\lesssim 5$) and for the 2.1 μm retrieval (relative to the 3.7 μm one). Thus, for thicker clouds and the 3.7 μm retrieval land surface albedo issues may be less problematic.

10 Finally, we note that the thermal emission correction for the MODIS $r_{e3.7}$ (see Section 3.1) retrieval has some uncertainty that should be considered; the uncertainty for this is included (combined with other uncertainties) in the MODIS Collection 6 pixel level uncertainty products (Platnick et al., 2017). It is possible that effects additional to those included, such as cloud heterogeneity, surface heterogeneity, etc., may further increase the uncertainty beyond that estimated in the MODIS products, but these are currently not well documented.

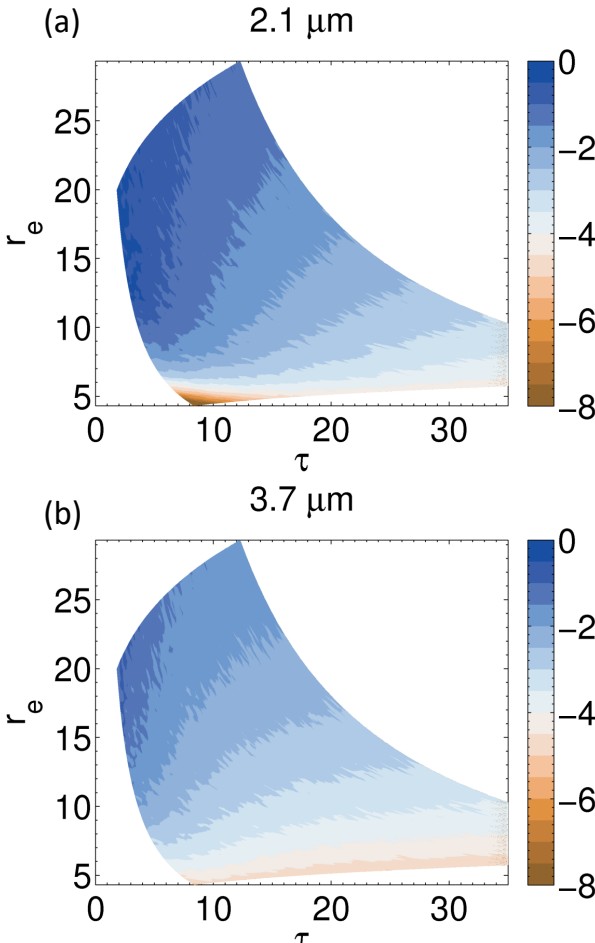

(a) 2.1 μm

(b) 3.7 μm

**Figure 10.** Percentage $\tau$ bias (retrieved minus actual value from the input model profile) as a function of $\tau$ and $r_e$.

## 6 Conclusions

We have described and quantified a positive bias in satellite retrievals of cloud droplet concentration ($N_d$) and Liquid Water Path (LWP) that make use of the adiabatic cloud assumption to estimate these quantities from satellite observed cloud optical depth ($\tau$), effective radius ($r_e$) and cloud top temperature. We term the bias the "vertical penetration bias". It arises due to the well–documented vertical penetration of photons with wavelengths in the shortwave-infrared range into the upper regions of clouds, so that $r_e$ retrievals are representative of values some distance below cloud top (Platnick, 2000; Bennartz and Rausch, 2017) rather than being those at cloud top as assumed by the $N_d$ and LWP retrievals. Here we quantified the optical depth as measured from cloud top downwards, $d\tau$, at which the retrieved $r_e$ equaled the actual $r_e$ for adiabatic clouds covering a large

range of total cloud optical depths and $N_d$ values. We showed that knowledge of $d\tau$ allows a corrected $N_d$ to be calculated by subtracting $d\tau$ from the observed $\tau$ and using that in the $N_d$ retrieval instead of $\tau$. We characterised $d\tau$ as functions of $\tau$ for the

2.1 and 3.7 μm $r_e$ retrievals ($r_{e2.1}$ and $r_{e1.6}$, respectively) and found that a 1-D relationship approximates the modelled data well. $d\tau$ increases with $\tau$ and is larger for $r_{e2.1}$ than for $r_{e3.7}$ and so the vertical penetration $N_d$ bias affects retrievals based on $r_{e2.1}$ more than those using $r_{e3.7}$. Similarly, we also parameterized the true cloud top effective radius ($r_e(H)$) as a function of the retrieved $r_e$ and $\tau$, allowing both a corrected $N_d$ and LWP to be calculated by using $r_e(H)$ instead of the retrieved value. Both the $d\tau$ and $r_e$ correction methods give similar results for the $N_d$ retrievals suggesting that the latter is preferable since it

also allows for a correction to LWP. However, for some applications it may be useful to be able to parameterize $d\tau$.

We quantified the vertical penetration $N_d$ bias for one–year $N_d$ and LWP data sets. The corrections presented here suggest that $N_d$ and LWP errors will increase as the $\tau$ value of the cloud scene gets lower. For many regions that are considered trustworthy for $N_d$ and LWP retrievals (typically stratocumulus regions), there are high frequencies of low $\tau$ values and the $N_d$ biases are significant. For example, for the SE Pacific and SE Atlantic regions clouds with $\tau \leq 10$ (for which $N_d$ errors are

expected to be $\geq 31\%$ for $r_{e2.1}$ and $\geq 15\%$ for $r_{e3.7}$) occur, respectively, 53 and 69% of the time on average. The mean $r_{e2.1}$ vertical penetration $N_d$ biases for these regions were 32 and 35%, respectively. Out of the stratocumulus regions examined, these two were the worst affected. For $r_{e3.7}$ the $N_d$ biases were much smaller; for example, mean biases for the SE Pacific and SE Atlantic regions were 15 and 17%, respectively. $N_d$ biases were predicted to be worse for the tropical and sub-tropical regions than for higher latitudes. The time-variability of the biases were also examined and were shown to be significant

(regional mean standard deviations of 19–37% and 32–56% for $r_{e2.1}$ and $r_{e3.7}$, respectively). This indicates that long term averages of the vertical penetration $N_d$ bias corrections are not useful for correcting $N_d$ data over short timescales (e.g. daily $N_d$ data). We also examined the seasonality of the $N_d$ biases and showed that, for the stratocumulus regions, generally the DJF season was worst affected, followed by SON.

LWP biases were of a lower magnitude than those for $N_d$ and were negative. The largest biases were again for the SE Atlantic

region where the mean bias was -11.1 % and the smallest for the Barents Sea region (-8 %). Biases were also lower when using $r_{e3.7}$ with a maximum (most negative) bias of -6.1 %.

We caution that the correction for the $N_d$ and LWP vertical penetration biases presented here should only be considered in combination with corrections for other biases that affect $\tau$ and $r_e$. Zhang et al. (2016) suggest a correction for the sub-pixel heterogeneity bias effect, but corrections may not currently exist for all biases and it is likely that some unidentified biases still exist. Therefore, we recommended that our correction is currently only applied to homogeneous cloud scenes in order to

minimize possible entanglements with biases resulting from heterogeneity effects, which are not accounted for. Such conditions can be obtained by limiting retrievals to associated heterogeneity ($H_\sigma$) values (available in MODIS MYD06 Collection 6 products) to less than about 0.1. Otherwise $N_d$ and LWP biases could be made worse; for example, in situations where the fortuitous cancellation of opposing errors leads to initially small $N_d$ errors. The latter was suspected to have occurred for the comparison between MODIS $N_d$ retrievals and in-situ aircraft observations as presented in Painemal and Zuidema (2011). We

showed that the SE Pacific, which is the region examined in that study, had a mean vertical penetration depth error of 32% suggesting that another unidentified $N_d$ bias may have been operating in order to give good agreement.

Previous studies have shown that $r_{e3.7}$ is less prone to biases due to sub-pixel averaging (Zhang and Plantnick, 2011; Zhang et al., 2012, 2016). Thus, combined with the work presented here, this supports the conclusion that $r_{e3.7}$ likely represents a better choice for use in $N_d$ and LWP retrievals.

For future work, it is recommended that additional characterization of $d\tau$ and $g_{re}$ is performed for a range of viewing geometries in order to ensure that the results presented here are robust for all cloud retrievals. The use of 3D radiative transfer calculations and simulated retrievals upon known LES model fields would also be useful for investigating how heterogenity

effects might interact with the vertical penetration effects. Further investigation into how the presence of precipitation affects our assumptions and results is also warranted.

## 7 Data availability

The data set is built from publically available MODIS Level-2 data (Collection 5). The MODIS data were obtained from NASA's Level 1 and Atmosphere Archive and Distribution System(LAADS http://ladsweb.nascom.nasa.gov/). NOAA_OI_SST_V2

data (SST data) was provided by the NOAA/OAR/ESRL PSD, Boulder, Colorado, USA, from their website at http://www.esrl.noaa.gov/psd/

*Author contributions.* D. P. Grosvenor developed the concepts and ideas for the direction of the paper, performed the error calculations using the MODIS data, produced the figures and wrote the manuscript. O. Sourdeval performed the retrievals upon the idealised adiabatic clouds. All authours provided additional input and comments on the paper during the paper writing process.

*Competing interests.* The authors declare that they have no conflict of interest.

*Acknowledgements.* DPG was funded by both the University of Leeds under Paul Field and from the NERC funded ACSIS programme via NCAS. OS was funded by the Federal Ministry for Education and Research in Germany (BMBF) in the High Definition Clouds and Precipitation for Climate Prediction (HD(CP)$^2$) project (FKZ 01LK1503A and 01LK1505E). RW's contribution was supported on NASA award number NNX16AP31G. The MODIS data were obtained from NASA's Level 1 and Atmosphere Archive and Distribution System(LAADS http://ladsweb.nascom.nasa.gov/). NOAA_OI_SST_V2 data (SST data) was provided by the NOAA/OAR/ESRL PSD, Boulder, Colorado,

USA, from their website at http://www.esrl.noaa.gov/psd/.

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
