# Peer review of "Quantifying and correcting the effect of vertical penetration assumptions on droplet concentration retrievals from passive satellite instruments"

_Atmospheric Measurement Techniques, 2017_

## Referee Comment (RC1) · Anonymous Referee #1 · 7 Feb 2018

Grosvenor et al. (2018) AMTD, REVIEW:

The manuscript describes a method that, in principle, corrects errors in adiabatic satellite cloud droplet number concentration (Nd) due to the inconsistency of utilizing satellite cloud effective radius (r_e) that represents values slightly below the cloud top, whereas satellite cloud optical depth (tau) fully captures the optical thickness of the clouds. To achieve this goal, the authors simulate a number of idealized cloud profiles with a 1D radiative transfer model, and then retrieve r_e and tau from the synthetic reflectances. Next, the authors derive an "effective" tau that corresponds to the optical

thickness where the retrieved $r_e$ and the synthetic $r_e$ match each other (the vertical penetration effect). They use the difference between the retrieved and the effective tau (applying a fit to their theoretical calculations) to quantify the error in MODIS-based Nd that does not account for the fact that the satellite $r_e$ is not exactly that at the cloud top due to the vertical photon penetration, which is in turn dependent on the sensor wavelength and the specific thickness of the cloud (and probably solar zenith angle and viewing geometry).

The manuscript makes an interesting use of the results in Platnick (2000), which shows that the retrieved $r_e$ should differ from the observed $r_e$ by a few um (or less) due to the photon penetration. The manuscript is concise and well-written, however when I first browsed the paper, I got confused about whether the authors wanted to show a real satellite bias in tau (and Nd) or a methodological bias (I realized it was the latter). My fundamental criticism of Grosvenor et al. is that, from a remote sensing point of view, the problem is not that the satellite tau should be reduced because $r_e$ is not at the cloud top. Instead, it is that $r_e$ is smaller than the observed $r_e$ at the top due to vertical stratification, and probably $r_e$ should be somehow increased (i.e. $r_e$ drives the uncertainty in Nd). This is the correct interpretation, as it is well known from the early work by Nakajima (King and co-authors) that satellite tau is almost insensitive to the cloud vertical structure, and only $r_e$ can be greatly affected by the vertical stratification. So, the Nd bias should be expressed in terms of a delta $r_e$. Another inconsistency (related to my previous comment) is with the use of the (pseudo) adiabatic model, which if I interpret correctly, it implies that the liquid water path (LWP) is proportional to $r_e$*tau. So, any error calculation applied to Nd has to be also valid for LWP. However, if we apply equation (13) to LWP, i.e.:

LWP_uncorrected/LWP_corrected=(tau/(tau-dtau))

Using a dtau=4.5 for tau=10 (figure 1a), then LWP_uncorrected/LWP_corrected=10/6.5=1.54. A 54% overestimation in LWP is clearly a mathematical contradiction. On the other hand, if, for instance, we utilize the results in Platnick (2000) for a cloud top $r_e$ =12

um, and a retrieved $r_e$= 10.7 um (2.1 um wavelength), we get:

LWP_uncorrected/LWP_corrected=r_uncorrected/r_corrected=10.7/12=0.89.

That is, the retrieved $r_e$ yields an underestimation of LWP. Again, this result points to a main reasoning problem in the manuscript, which is, the error should not be expressed in terms of tau.

Lastly, the authors say that there are several other errors that can bias $r_e$ and tau. This is a key statement, and a literature review will show that biases in $r_e$ are not dominated by the cloud vertical stratification (I am not aware of any studies that actually show an adiabatic signature in the satellite $r_e$ bias). For instance, If one calculates the difference between MODIS $r_e$ at 2.1 um and 3.7 um, the difference is positive everywhere over the ocean (the difference can be larger than 5 um, see Fig. 10 in Zhang and Platnick, 2011). This result suggests that the error discussed in Grosvenor et al. is negligible. So, I find it surprising that the authors found errors up to 50 % in Nd (Figure 6), which is very large. Since their results are only valid in a plane parallel world (sub-pixel variability is not accounted for) and with the use of idealized profiles, the validity of the correction cannot be demonstrated. The authors do discuss some of these issues but, unfortunately, the main concern remains, that is, it is unclear that the correction will yield an improved estimate of cloud droplet number concentration.

---

## Referee Comment (RC2) · Z. Zhang (Referee) · 8 Mar 2018

Z. Zhang (Referee)

zhibo.zhang@umbc.edu

Review on "Quantifying and correcting the effect of vertical penetration assumptions on droplet concentration retrievals from passive satellite instruments" by Grosvenor et al.

Summary: This paper assesses a particular type of error in the satellite-based retrieval of cloud droplet number concentration (Nd) retrieval from passive sensors such as MODIS. The error stems from the fact the shortwave infrared band used in MODIS

cloud droplet size retrieval (i.e., cloud droplet effective radius CER) does not correspond to the CER at the exact cloud top, but somewhere below the cloud top due to the penetration of the light into the cloud (termed as the "penetration depth bias" in this study), which leads to underestimation of CER and overestimation of Nd. This study investigates the size of this bias under different conditions and also provide a simple parameterization scheme to correct this bias in the observation.

The topic of this paper is suitable for AMT. The paper is well-written, concise and easy to follow for the readers with the right background (but perhaps too technical for general readers). Overall, I recommend publication after some revision.

Comments/Suggestions:

My biggest concern/criticism for this study and many other studies on Nd retrieval is that most of them are based on highly idealized cloud model, namely, the perfect, 1D, plane-parallel, adiabatic cloud with linear LWC lapse rate and constant Nd. It seems to me that, the meaningfulness of this study depends pretty much on the validity of this ideal cloud model. In particular, it is well known that the entrainment process can significantly affect the cloud microphysics at cloud top and thereby deviates the cloud vertical structure from the classic model assumed in Nd retrieval. How may the cloud top entrainment process influence those equations in section 2? What is the typical vertical scale of cloud top entrainment in comparison with the penetration depth of the SWIR band? Do homogenous mixing and inhomogeneous mixing as a result of cloud top entrainment have a different or similar impact on cloud top CER structure and Nd retrieval? At least, these questions should be mentioned, discussed with some references.

What is the COT ($\tau$) used in the Nd retrieval? Note that in MODIS operational retrieval, clouds are assumed to be vertically homogeneous. Because of the "penetration depth bias", the retrieve CER is different from the CER at the cloud top. Another possible bias is that the retrieved COT is different from the true COT. This might be small but

should be quantified.

In this study, only the solar reflective part of the 3.7 $\mu$m band is considered. In reality, the radiance in this band is contributed by two parts during the daytime, the solar reflection and thermal emission. The emission part is "corrected" based on the 11 $\mu$m band radiance in the MODIS retrieval. This should be pointed out and if the correction process could somehow confound the results then some discussion is needed. This is especially important as the paper claims that 3.7 $\mu$m band is better for Nd retrieval (which I agree) than the 2.1 $\mu$m band.

For 3.7 $\mu$m band, its weighting function is close to two-way transmittance. I'd like to encourage the author to try to come up with an analytical solution of CER* if the weighting function follows the two-way transmittance. A paper that might be helpful Zhang et al. 2017 JGR (http://onlinelibrary.wiley.com/doi/10.1002/2016JD025763/full) (Equation 4)

Why is land always masked in Nd retrievals? Why or why not can the same method be applied to land?

---

## Author Response (AR1)

We thank both of the reviewers for taking the time to read and comment on the paper; your comments have helped to greatly improve the paper. The reviewer comments are repeated below in green italics and our responses are in black.

**Responses to Reviewer 1**

*The manuscript describes a method that, in principle, corrects errors in adiabatic satellite cloud droplet number concentration (Nd) due to the inconsistency of utilizing satellite cloud effective radius (r_e) that represents values slightly below the cloud top, whereas satellite cloud optical depth (tau) fully captures the optical thickness of the clouds. To achieve this goal, the authors simulate a number of idealized cloud profiles with a 1D radiative transfer model, and then retrieve r_e and tau from the synthetic reflectances. Next, the authors derive an "effective" tau that corresponds to the optical thickness where the retrieved r_e and the synthetic r_e match each other (the vertical penetration effect). They use the difference between the retrieved and the effective tau (applying a fit to their theoretical calculations) to quantify the error in MODIS-based Nd that does not account for the fact that the satellite r_e is not exactly that at the cloud top due to the vertical photon penetration, which is in turn dependent on the sensor wavelength and the specific thickness of the cloud (and probably solar zenith angle and viewing geometry).*

*The manuscript makes an interesting use of the results in Platnick (2000), which shows that the retrieved r_e should differ from the observed r_e by a few um (or less) due to the photon penetration. The manuscript is concise and well-written, however when I first browsed the paper, I got confused about whether the authors wanted to show a real satellite bias in tau (and Nd) or a methodological bias (I realized it was the latter).*

*My fundamental criticism of Grosvenor et al. is that, from a remote sensing point of view, the problem is not that the satellite tau should be reduced because r_e is not at the cloud top. Instead, it is that r_e is smaller than the observed r_e at the top due to vertical stratification, and probably r_e should be somehow increased (i.e. r_e drives the uncertainty in Nd). This is the correct interpretation, as it is well known from the early work by Nakajima (King and co-authors) that satellite tau is almost insensitive to the cloud vertical structure, and only r_e can be greatly affected by the vertical stratification. So, the Nd bias should be expressed in terms of a delta r_e. Another inconsistency (related to my previous comment) is with the use of the (pseudo) adiabatic model, which if I interpret correctly, it implies that the liquid water path (LWP) is proportional to r_e*tau.*
*So, any error calculation applied to Nd has to be also valid for LWP.*

*However, if we apply equation (13) to LWP, i.e.:*

*LWP_uncorrected/LWP_corrected=(tau/(tau-dtau))*

*Using a dtau=4.5 for tau=10 (figure 1a), then LWP_uncorrected/LWP_corrected=10/6.5=1.54. A 54% overestimation in LWP is clearly a mathematical contradiction. On the other hand, if, for instance, we utilize the results in Platnick (2000) for a cloud top r_e =12 um, and a retrieved r_e= 10.7 um (2.1 um wavelength), we get:*

*LWP_uncorrected/LWP_corrected=r_uncorrected/r_corrected=10.7/12=0.89.*

*That is, the retrieved r_e yields an underestimation of LWP. Again, this result points to a main reasoning problem in the manuscript, which is, the error should not be expressed in terms of tau.*

We were originally considering the special case of the Nd retrieval that makes the assumption that r_e is at cloud top and the application to LWP was not considered; the original method using the tau correction only applies to the Nd retrieval and not to the LWP calculation, which we did not make clear. We originally chose to use a tau correction since the correction for tau seemed simpler to parameterize and less prone to uncertainties in the parameterization. Further work following from your suggestions (and those of the other reviewer) has shown that dre/re can also be modelled fairly accurately with a fitted curve and so we now also present results using this formulation and make some estimates of the LWP bias too. The following new figures are included showing 2D histograms of the re correction (divided by re)vs the optical depth, along with the parameterized fit to the data (as previously shown for the tau correction) :-

[Figure]

*Lastly, the authors say that there are several other errors that can bias r_e and tau. This is a key statement, and a literature review will show that biases in r_e are not dominated by the cloud vertical stratification (I am not aware of any studies that actually show an adiabatic signature in the satellite r_e bias). For instance, If one calculates the difference between MODIS r_e at 2.1 um and 3.7 um, the difference is positive everywhere over the ocean (the difference can be larger than 5 um, see Fig. 10 in*

*Zhang and Platnick, 2011). This result suggests that the error discussed in Grosvenor et al. is negligible. So, I find it surprising that the authors found errors up to 50 % in Nd (Figure 6), which is very large. Since their results are only valid in a plane parallel world (sub-pixel variability is not accounted for) and with the use of idealized profiles, the validity of the correction cannot be demonstrated. The authors do discuss some of these issues but, unfortunately, the main concern remains, that is, it is unclear that the correction will yield an improved estimate of cloud droplet number concentration.*

We agree that other r_e biases are important for Nd retrievals and are probably of equal or stronger magnitude than the changes in r_e due to the vertical profile changes. In the original paper we wrote (p. 16, line 10) :-

- "It is also clear that the suggested correction for the vertical penetration effect should only be applied to the retrievals of Nd with consideration of other bias sources. These other potential error sources are numerous and include re biases due to sub-pixel heterogeneity (Zhang and Platnick, 2011; Zhang et al., 2012, 2016); 3D radiative effects (Marshak et al., 2006);…" (etc.)

And on p.10, line 12:-

- "Thus, the application of the correction described in this paper in isolation has the potential to enhance any negative bias in Nd caused by a positive re bias."

we caution the reader that the bias correction should only be applied after other biases have been accounted for. We realize that this limits the current usefulness of the correction until the other biases have been quantified - we therefore have added the following text to the and of the above :-

- "; it is thus recommended that the bias correction is not applied until the other error sources have been fully characterized."

Also we have modified the following paragraph in the conclusions to make this point clearer and to recommend restriction to low heterogeneity situations (see later for the justification for this) :-

We caution that the correction for the $N_d$ and LWP vertical penetration biases presented here should only be considered in combination with corrections for other biases that affect $\tau$ and $r_e$. Zhang et al. (2016) suggest a correction for the sub-pixel heterogeneity bias effect, but corrections may not currently exist for all biases and it is likely that some unidentified biases still exist. Therefore, we recommended that our correction is currently only applied to homogeneous cloud scenes in order to minimize possible entanglements with biases resulting from heterogeneity effects, which are not accounted for. Such conditions can be obtained by limiting retrievals to associated heterogeneity ($H_\sigma$) values (available in MODIS MYD06 Collection 6 products) to less than about 0.1. Otherwise $N_d$ and LWP biases could be made worse; for example, in situations where the fortuitous cancellation of opposing errors leads to initially small $N_d$ errors. The latter was suspected to have occurred for the comparison between MODIS $N_d$ retrievals and in-situ aircraft observations as presented in Painemal and Zuidema (2011). We showed that the SE Pacific, which is the region examined in that study, had a mean vertical penetration depth error of 35% suggesting that another unidentified $N_d$ bias may have been operating in order to give good agreement.

However, we feel that it is useful and important to quantify the vertical penetration bias nevertheless and to suggest ways to remedy it (albeit in the sense of an idealised retrieval with no other bias sources). The addition of a previously unconsidered underlying bias is important since, for example,  it

would disrupt the cancellation of the other errors that led to the good agreement between aircraft and MODIS Nd seen in Painemal and Zuidema (2012), possibly suggesting that another unaccounted for error source exists.

The question is whether the vertical penetration effect occurs in addition to the other errors; e.g., whether (scenario A) the presence of cloud heterogeneity somehow prevents the effects of the vertical stratification from influencing the retrieved r_e and makes it irrelevant, or whether (scenario B) the vertical stratification is influencing r_e in the expected way (i.e. a tendency to cause re_37>re_21), but with a corresponding counter-influence in the opposite direction due to heterogeneity. We argue for scenario B, but it is hard to prove this within the scope of this study, since it would likely require computationally expensive 3D radiative transfer modelling of known cloud fields (e.g., from LES models), followed by r_e and tau retrievals.

Some evidence for scenario B is that it may explain why VOCALS aircraft measurements showed that re_21 and re_37 were very similar; it is possible that sub-pixel (or other) heterogeneity effects tended to increase re_21 relative to re_37, but that the vertical penetration effect has the opposite tendency, resulting in overall similar values. We also note that there are many situations when the expected vertical stratification of r_e does occur (i.e. re_37>re_21), as demonstrated in the following figure (included in the revised paper):-

[Figure]

It shows the percentage of pixels where re_37>re_21 for 90 days (Jan, Feb, Mar) of 0.1° resolution Collection 6 MODIS observations (single layer liquid clouds only; filtered to exclude tau<5 and partially cloudy pixels). The four panels are for four different bins of the heterogeneity parameter (the standard deviation of the 250m resolution 0.86um reflectance divided by the mean reflectance) with bin ranges labelled above the panels. It is clear that for many regions the relative r_e values that are consistent

with an adiabatic profile occur more than 50% of the time, particularly when the cloud heterogeneity is low.  Similarly, the Bennartz (2017) Nd dataset requires that re_37>re_21 in order for a datapoint to be included in the dataset indicating that there are a lot of times when this is the case.

We have also added this figure, which shows the ratio between re_37 and re_21 for an example MODIS scene :-

[Figure]

It shows that the ratio is generally larger than one for the standard uncorrected MODIS product values (left) for the overcast stratocumulus regions. The right plot shows the ratios after the effective radius correction has been applied (based on our parameterizations for both 2.1 and 3.7um). If our parameterization is working well and if the re_21 vs re_37 differences are caused by vertical stratification then the ratio should equal one. It can be seen that this is indeed the case for most of the overcast region with just a small amount of positive bias in the overcast clouds to the west and north. The more broken cloud regions should be ignored since Nd retrievals would ideally not be made for such clouds.

Some discussion on these issues has been added to the Discussion section of the revised paper :-

Our paper quantifies the vertical penetration bias in isolation to the other effects mentioned above. It should be questioned, though, whether the presence of cloud heterogeneity and other effects somehow prevent the effects of the vertical stratification from influencing the retrieved $r_e$, making it irrelevant. This could be a potential explanation for why it is often observed that $r_{e2.1}$ is larger than $r_{e3.7}$ (Zhang and Plantnick, 2011) in contrast to the direction expected from adiabatic clouds given the vertical penetration effect, since it is known that sub-pixel heterogeneity effects tend to cause a positive $r_{e2.1}$ bias relative to $r_{e3.7}$ (Zhang et al., 2012). We argue, though, that the vertical stratification effect occurs in addition to other effects (heterogeneity, etc.) with the latter cancelling out and often exceeding the former such that the positive difference between $r_{e2.1}$ and $r_{e3.7}$ would be even larger without the vertical stratfication effect. The cancellation of biases may also explain why VOCALS aircraft measurements (Painemal and Zuidema, 2011) tended to show that $r_{e2.1}$ and $r_{e3.7}$ were very similar. We also note that there are many situations when the expected result due to vertical stratification of $r_e$ does occur (i.e. $r_{e3.7} > r_{e2.1}$), as demonstrated in Fig. 8. This shows ratios between $r_{e3.7}$ and $r_{e2.1}$ for an example MODIS scene in the SE Pacific stratocumulus region. Ratios using the uncorrected MODIS $r_e$ values are shown, which shows that the ratio is exceeds one for most of the stratocumulus cloud region (the clouds that adjoin the coast) with ratios ranging from around 1.1 to 1.2. In the more broken clouds the ratio is less than one, which is likely a result of cloud heterogeneity. However, it would be expected that $N_d$ retrievals would not be applied to such clouds. The figure also shows the ratios calculated using $r_{e3.7}$ and $r_{e2.1}$ values that have been corrected using the $g_{re}$ factors. If the differences between $r_{e3.7}$ and $r_{e2.1}$ were caused by vertical stratification alone and if our parameterization was correctly predicting the cloud top $r_e$ for both MODIS channles then this ratio should be equal to one. This is the case for the clouds close to the coast indicating that our parameterization is working well for these clouds. The ratios are a little higher than one further north and west (around 1.05-1.08) indicating that either our parameterization is not working correctly for these clouds, or that other factors are causing relative differences between $r_{e3.7}$ and $r_{e2.1}$ (e.g. sub-adiabaticity, cloud heterogeneity, etc.). Figure 9 shows the percentage of pixels for which $r_{e3.7} > r_{e2.1}$ for 90 days of MODIS SE Pacific observations divided into four different heterogeneity bins. Heterogeneity is characterized by the $H_\sigma$ parameter (Liang et al., 2009), which is the standard deviation of the 250 m resolution 0.86 μm reflectance ($R_{0.86}$) divided by the mean $R_{0.86}$. It is clear that for many regions relative $r_e$ values that are consistent with an adiabatic profile occur more than 50% of the time, particularly when the cloud heterogeneity is low. This suggests that it may be possible to use $H_\sigma$ to determine the situations in which the bias correction is more applicable. However, it is hard to definitively prove our argument within the scope of this study, particularly for more heterogeneous regions, since it would likely require computationally expensive 3D radiative transfer modelling of known cloud fields (e.g., from LES models), followed by $r_e$ and $\tau$ retrievals.

**References**

Bennartz, R. and Rausch, J.: Global and regional estimates of warm cloud droplet number concentration based on 13 years of AQUA-MODIS observations, Atmospheric Chemistry and Physics, 17, 9815–9836, doi:10.5194/acp-17-9815-2017, https://doi.org/10.5194/acp-17-9815-2017, 2017.

**Responses to Reviewer 2**

- *Summary:*

    *This paper assesses a particular type of error in the satellite-based retrieval of cloud droplet number concentration (Nd) retrieval from passive sensors such as MODIS. The error stems from the fact the shortwave infrared band used in MODIS cloud droplet size retrieval (i.e., cloud droplet effective radius CER) does not correspond to the CER at the exact cloud top, but somewhere below the cloud top due to the penetration of the light into the cloud (termed as the "penetration depth bias" in this study), which leads to underestimation of CER and overestimation of Nd. This study investigates the size of this bias under different conditions and also provide a simple parameterization scheme to correct this bias in the observation. The topic of this paper is suitable for AMT. The paper is well-written, concise and easy to follow for the readers with the right background (but perhaps too technical for general readers). Overall, I recommend publication after some revision.*

- *Comments/Suggestions:*

    *My biggest concern/criticism for this study and many other studies on Nd retrieval is that most of them are based on highly idealized cloud model, namely, the perfect, 1D, plane-parallel, adiabatic cloud with linear LWC lapse rate and constant Nd. It seems to me that, the meaningfulness of this study depends pretty much on the validity of this ideal cloud model.*

    *In particular, it is well known that the entrainment process can significantly affect the cloud microphysics at cloud top and thereby deviates the cloud vertical structure from the classic model assumed in Nd retrieval. How may the cloud top entrainment process influence those equations in section 2? What is the typical vertical scale of cloud top entrainment in comparison with the penetration depth of the SWIR band? Do homogenous mixing and inhomogeneous mixing as a result of cloud top entrainment have a different or similar impact on cloud top CER structure and Nd retrieval?At least, these questions should be mentioned, discussed with some references.*

We have now included some discussion on entrainment effects in the Discussion section. The evidence suggests that for stratocumulus clouds cloud top entrainment results in extreme inhomogeneous mixing, so that the CER remains constant - this is also backed up by the VOCALS aircraft measurements. The other effects due to the non-constant Nd profile and non-adiabatic liquid water content profiles are likely to be small since we estimate from aircraft observations that the entrainment region only contributes around 0.5 optical depths to the total optical depth. The added text is as follows :-

Evaporation effects related to entrainment also have the potential to reduce $r_e$, $N_d$ and $LWC$ near cloud top and hence negate some of the assumptions upon which the $N_d$ retrievals rest. However, we argue that the entrainment effect upon $r_e$ is likely to be minimal for two reasons: Firstly, the evidence suggests that for stratocumulus clouds extreme inhomogeneous mixing occurs at cloud top, which reduces the $LWC$ and $N_d$, but does not change $r_e$ (Burnet and Brenguier, 2007; Brenguier et al., 2011; Painemal and Zuidema, 2011). Secondly, the results of Painemal and Zuidema (2011) indicate that entrainment occurs within approximately the first 0.5 optical depths from cloud top on average; the penetration depths calculated here are considerably

larger than this for reasonably thick clouds (Fig. 2). The effect of the reduced $N_d$ and $LWC$ within the entrainment zone is not so clear-cut; this would negate the assumption of a vertically constant $N_d$ and monotonically increasing $LWC$ used to formulate the total $\tau$. However, given the likely small $\tau$ contribution from the entrainment region relative to the total $\tau$, this effect is likely to be small.

*What is the COT ($\tau$ ) used in the Nd retrieval? Note that in MODIS operational retrieval, clouds are assumed to be vertically homogeneous. Because of the "penetration depth bias", the retrieve CER is different from the CER at the cloud top. Another possible bias is that the retrieved COT is different from the true COT. This might be small but should be quantified.*

The COT used in our Nd retrieval is that directly from the MODIS products and so may contain biases due to the non-uniform CER profile that is likely to occur in reality combined with the fact that MODIS assumes vertically uniform clouds. We have now included a figure that quantifies the percentage bias in the retrieved optical depth (relative to the model profile optical depth) :-

[Figure]

**Figure 8.** Percentage $\tau$ bias (retrieved minus actual value from the input model profile) as a function of $\tau$ and $r_e$.

The following text has also been added to the discussion :-

The MODIS retrieval uses reflectances from both a visible and a shortwave infra-red (SWIR) wavelength channel with the former being primarily determined by $\tau$ and the latter by $r_e$. However, a bi-spectral retrieval is used and so there is also some sensitivity of the retrieved $\tau$ to the SWIR reflectance, which will be representative of the $r_e$ below cloud top due to the vertical penetration effect. This, combined with the fact that the MODIS forward retrieval model assumes a vertically uniform cloud, will result in the retrieved $\tau$ being biased relative to the real value (assuming the real cloud has an adiabatic profile). Figure 9 shows the difference between the retrieved and model profile $\tau$; the bias is negative and smaller in magnitude than 5 % for the 3.7 μm retrieval. They are slightly larger for the 2.1 μm retrieval, but still lower in magnitude than 5 %, except at $r_e \lesssim 7$ μm. Although it should be noted that some of this bias may be due to other causes related to the inconsistencies between the vertically uniform and adiabatic models, rather than the $r_e$ vertical penetration bias. Since the retrieved $N_d$ is proportional to the square root of $\tau$, this will lead to small $N_d$ biases. Biases in LWP will be similar to those in $\tau$ since LWP is proportional to $\tau$, but $r_e$ biases are still likely to dominate (e.g., see Fig. 1). Thus, we have not pursued this further.

*In this study, only the solar reflective part of the 3.7 μm band is considered. In reality, the radiance in this band is contributed by two parts during the daytime, the solar reflection and thermal emission. The emission part is "corrected" based on the 11 μm band radiance in the MODIS retrieval. This should be pointed out and if the correction process could somehow confound the results then some discussion is needed. This is especially important as the paper claims that 3.7 μm band is better for Nd retrieval (which I agree) than the 2.1 μm band.*

We have added some discussion on this in the methods and now describe how this is dealt with in our retrievals :-

and 3.7μm wavelengths, which are hereafter referred to to as $r_{e2.1}$ and $r_{e3.7}$, respectively. The MODIS $r_{e3.7}$ retrieval requires a correction to account for the contribution to the observed radiance from thermal emission, which is based on the observed 11 μm radiance (Platnick and Valero, 1995; King et al., 2015; Platnick et al., 2017). We account for this in our retrievals by removing the thermal contribution during the RT calculation instead of via the 11 μm radiance, which should produce a consistent end result. The RT calculations were performed assuming a black surface, a clear atmosphere (i.e. gaseous absorption is neglected),

And we also add some discussion in the Discussion section :-

Finally, we note that the thermal emission correction for the MODIS $r_{e3.7}$ (see Section 3.1) retrieval has some uncertainty that should be considered; the uncertainty for this is included (combined with other uncertainties) in the MODIS Collection 6 pixel level uncertainty products (Platnick et al., 2017). It is possible that effects additional to those included, such as cloud heterogeneity, surface heterogeneity, etc., may further increase the uncertainty beyond that estimated in the MODIS products, but these are currently not well documented.

*For 3.7 μm band, its weighting function is close to two-way transmittance. I'd like to encourage the author to try to come up with an analytical solution of CER\* if the weighting function follows the two-way transmittance.*
*A paper that might be helpful Zhang et al. 2017 JGR*
*(http://onlinelibrary.wiley.com/doi/10.1002/2016JD025763/full) (Equation 4)*

We have now been able to parameterize the correction to CER in order to return the cloud top CER as a function of the retrieved optical depth and retrieved CER. This was also suggested by Reviewer 1 - please see the response given there for details.

*Why is land always masked in Nd retrievals? Why or why not can the same method be applied to land?*

We restricted the analysis to ocean retrievals since the Nd and LWP retrievals are most suited to stratocumulus and these types of clouds occur much more frequently over the oceans. Land retrievals also add additional complications from surface albedo uncertainties and cloud masking problems, which we also tried to avoid. We have added some discussion on these points :-

In this paper we have only considered retrievals over the ocean, although retrievals over land for $\tau$ and $r_e$ are available for MODIS. MODIS surface albedo uncertainties are likely to be much higher over land than over the oceans (King et al., 2004; Rosenfeld et al., 2004; Bréon and Doutriaux-Boucher, 2005) since the surface albedo is much more variable over land. In addition, cloud masking is more difficult over land, particularly over non–vegetated surfaces, transitional areas between desert and vegetated surfaces and above high-altitude regions (Platnick et al., 2003). We have ignored land regions in order to avoid such complications and also because stratocumulus clouds are more prevalent over ocean regions (Klein and Hartmann, 1993; Wood, 2012). However, the results shown in this paper may still apply over land. The results of Rosenfeld et al. (2004) and Platnick et al. (2017, their Fig. 14) suggest that surface albedo uncertainties are more important at lower optical depths ($\lesssim 5$) and for the 2.1 μm retrieval (relative to the 3.7 μm one). Thus, for thicker clouds and the 3.7 μm retrieval land surface albedo issues may be less problematic.

**List of changes**

- A parameterization of the corrected effective radius (reff) as a function of the retrieved reff and optical depth has been added.
    - A new figure for this has been added, along with a new equation for the parameterized fit and a new table listing the fit coefficients. Also, a new equation showing the ratio of uncorrected to corrected droplet concentrations as a function of the correction parameter.
    - Additional text has been added to describe this in the Abstract, Section 2, Section 3.1 and the conclusions.
- The new parameterization is now used in place of the old one, which makes small differences to the results. Figure 4 (of new manuscript) is also updated using the new parameterization.
- The new parameterization allows LWP biases to also be estimated and so new text describing this has been added to the Abstract, Section 2, Section 4 (Results) and Conclusions
- The previous Table 2 has been split into two in order to separate the regional optical depth statistics from the biases. The LWP biases are now also listed.
- Following these changes the title has been changed to :-
    - "Parameterizing cloud top effective radii from satellite retrieved values, accounting for vertical photon transport: Quantification and correction of the resulting bias in droplet concentration and liquid water path retrievals."
- Information on the thermal emission correction for 3.7 micron MODIS reff retrievals has been added to Section 3.1.
- The data from Figure 3 now includes the tau>5 filtering for consistency with the rest of the paper.
- Several additional discussion points have been added to the Discussion section following the reviewer comments (see the responses for a point-by-point response and more details on the additions), including paragraphs on :-
    - The effects of entrainment on reff retrievals and our results.
    - Whether other effects (such as heterogeneity) prevent the vertical stratification from having an impact. Two new figures have been added to aid with this (Figs. 8 and 9) and are discussed. This leads to the recommendation to restrict the use of the bias correction to more homogeneous clouds in the conclusions.
    - How much optical depth retrievals are likely to be biased due to vertical penetration effects (including a new figure – Fig. 10 - to quantify this).
    - Additional biases that are likely to occur over land as justification for not including land retrievals.
    - The thermal correction for the 3.7 micron reff retrieval.
- A few grammatical changes have also been made.

[revised manuscript text omitted]

---

## Author Response (AR2)

**Responses to reviewer comments for Grosvenor et al., AMT, 2018.**

We thank the reviewer for their further helpful comments and address the additional points below.

**1) The new uncertainty discussion is also very informative. I am wondering if the use of a constant cw=1.81*10-6 is an additional source of error (keep in mind that in the extra-tropics and for supercooled clouds, cw should be smaller than 1.81*10-6).**

*Thanks for the suggestion. We have tested the effect of cw on the estimated values of dtau and g_re and included the results (for the mean values in each tau bin) in the revised text :-*

In addition, a liquid water condensation rate ($c_w$) value of $1.81 \times 10^{-6}$ kg m$^{-4}$ was assumed for the model adiabatic clouds, which corresponds to a cloud temperature of 278 K at a pressure of 850 hPa. In reality, cloud temperatures and hence $c_w$ will vary mainly as a function of cloud temperature. We have performed sensitivity tests using a value of $1.0 \times 10^{-6}$ kg m$^{-4}$, which corresponds to a cloud temperature of 262 K. This is likely to be close to the coldest temperature attained by boundary layer clouds over the oceans that are coupled to the surface, which are generally the types of clouds for which the droplet concentration retrievals are applied. The results (not shown) reveal mean differences (across the $\tau$ values tested) in the mean $d\tau$ line (i.e., the white line in Fig. 2a) of 4.2 % (the maximum difference was 10 %) for the 2.1 μm retrieval and 3.6 % (maximum of 7.6 %) for the 3.7 μm one. For $g_{re}$ the differences were of the opposite sign and much smaller, with mean differences in the median $d\tau$ line of -0.45 % (the most negative difference was -0.6 %) for the 2.1 μm retrieval and -0.3 %

(-0.66 % was the most negative difference) for the 3.7 μm one. Therefore, the effect of $c_w$ changes is relatively minor. These results would also apply for equivalent changes in the cloud adiabaticity (i.e., the value of $f_{ad}$).

**2) Similarly, the fact that the simulations utilize a constant cloud thickness = 1km is inadvertently shaping the vertical profile of re and extinction. Let's say that a typical cloud thickness for stratocumulus clouds is 400 m, this implies that the cloud top re is smaller than that for a 1000 thickness clouds for the same Nd (i.e., the cloud base/top gradient in r_e would decrease, which could have implications for the photon penetration). I understand that some simplifications need to be made, but I am wondering if something can be said about the dependence of the calculations on the physical thickness.**

*I think there may a slight mis-understanding here. We do not assume a 1km depth for the model adiabatic clouds upon which the retrievals are performed to get dtau and g_re; the depth of these clouds is determined by the LWP and cw value. We only assume a 1km depth for the plane parallel clouds that*

*are used to create look-up tables for the reflectances of cloud of specified tau and reff (in order to mimic MODIS-like retrievals). The depth of these clouds is arbitrary and similar assumptions are also made for operational retrievals (e.g., MODIS).*

**3) Figure 6: do the figures show the standard deviation of the bias relative to the mean bias? (the caption is a little bit difficult to understand). If so, it would be more informative to express the bias variability relative to the Nd variability, i.e. STD(Ndtrue-Ndretrieved)/STD(Ndtrue).**

*Figure 6 shows sig(B_prc)/mean(B_prc), where B_prc are the percentage Nd biases. The idea here was to give an idea about how accurate it would be to use the time mean percentage bias to correct the Nd values instead of a correction that was calculated daily. The caption has been updated accordingly :-*

**Figure 6.** As for Fig. 5 except showing the relative (as a percentage) standard deviation of the percentage $N_d$ bias over time, i.e. $\sigma_{\%bias}/\overline{\%bias}$.

**4) I agree with the authors that the correction could be applied to spatially homogeneous clouds. Consistent with Figure 9, Painemal&Minnis&Sun-Mack (2013, ACP) show that for homogeneous scenes and independent of their total liquid water path (estimated using a microwave satellite sensor), 3.7um re > 2.1 um re. This also suggests that entrainment and precipitation have a secondary role in explaining differences between 3.7 and 2.1um re (at least for stratocumulus clouds).**

*Thanks – we have added a reference to Painemal, Minnis and Sun-Mack in the discussion :-*

measurements (Painemal and Zuidema, 2011) tended to show that $r_{e2.1}$ and $r_{e3.7}$ were very similar. We also note that there are many situations when the expected result due to vertical stratification of $r_e$ does occur (i.e. $r_{e3.7} > r_{e2.1}$), as demonstrated in Painemal et al. (2013) and in Figure 8. This shows the percentage of pixels for which $r_{e3.7} > r_{e2.1}$ for 90 days of MODIS SE

**5) Abstract: I do not think VOCALS should be mentioned in the abstract (VOCALS data are not used in the article). Moreover, the correction technique is not limited to the VOCALS domain."**

*Thanks, we have removed the reference to VOCALS in the abstract.*

[revised manuscript text omitted]